# Contrasting seismic risk for Santiago, Chile, from near-field and distant earthquake sources

Ekbal Hussain[1,2], John R. Elliott[1], Vitor Silva[3], Mabé Vilar-Vega[3], and Deborah Kane[4]

[1]COMET, School of Earth and Environment, University of Leeds, LS2 9JT, UK
[2]British Geological Survey, Natural Environment Research Council, Environmental Science Centre, Keyworth, Nottingham, NG12 5GG, UK
[3]GEM Foundation, Via Ferrata 1, 27100 Pavia, Italy
[4]Risk Management Solutions, Inc., Newark, CA, USA

*Correspondence to:* Ekbal Hussain (ekhuss@bgs.ac.uk)

**Abstract.** More than half of all the people in the world now live in dense urban centres. The rapid expansion of cities, particularly in low-income nations, has enabled the economic and social development of millions of people. However, many of these cities are located near active tectonic faults that have not produced an earthquake in recent memory, raising the risk of losing the hard-earned progress through a devastating earthquake. In this paper we explore the possible impact that earthquakes can pose to the city of Santiago in Chile from various potential near-field and distant earthquake sources. We use high resolution stereo satellite imagery and derived digital elevation models to accurately map the trace of the San Ramón Fault, a recently recognised active fault located along the eastern margins of the city. We use scenario based seismic risk analysis to compare and contrast the estimated damage and losses to the city from several potential earthquake sources and one past event, comprising i) rupture of the San Ramón Fault, ii) a hypothesised buried shallow fault beneath the centre of the city, iii) a deep intra-slab fault, and iv) the 2010 Mw 8.8 Maule earthquake. We find that there is a strong magnitude-distance trade-off in terms of damage and losses to the city, with smaller magnitude earthquakes on more local faults, in the magnitude range 6-7.5, producing 9 to 17 times more damage to the city and estimated fatalities compared to the great magnitude 8+ earthquakes located offshore on the subduction zone. Our calculations for this part of Chile show that unreinforced masonry structures are the most vulnerable to these types of earthquake shaking. We identify particularly vulnerable districts, such as Ñuñoa, Santiago and Providencia, where targeted retrofitting campaigns would be most effective at reducing potential economic and human losses. Due to the potency of near-field earthquake sources demonstrated here, our work highlights the importance of also identifying and considering proximal minor active faults for cities in seismic zones globally, in addition to the more major distant large fault zones that are typically focused on in the assessment of hazard.

# 1 Introduction

Earthquakes are caused by the sudden release of accumulated tectonic strain that increases in the crust over decades to millennia. Many faults are often not recognised as dangerous because they have not recorded an earthquake in living and written memory (e.g. England and Jackson, 2011), and since probabilistic seismic hazard assessments (PSHA) rely on knowledge of past seismicity to determine hazard levels, the regions around these faults are often deemed to be low hazard in seismic risk assessments; until an earthquake strikes and the assessment is revised (Stein et al., 2012). The Mw 7.0 2010 Haiti earthquake, with its close proximity to an urban centre, was a stark reminder of how ruptures on these faults can be so deadly, especially when they are located near major population centres in poorly prepared low-income nations (Bilham, 2010).

The South American country of Chile is one of the most seismically active countries in the world. Since 1900 there have been 11 great earthquakes in the country with magnitudes 8 or larger (USGS, 2018). All of these were located on or near the subduction interface where the Nazca plate is subducting beneath the South American plate at 7.5 cm yr$^{-1}$ (DeMets et al., 2010), giving rise to the Andean mountain range (Armijo et al., 2015; Oncken et al., 2006). It is therefore unsurprising that shaking from offshore subduction zone events dominate the seismic hazard and thus the building design criteria in Chile (e.g. Fischer et al., 2002; Pina et al., 2012; Santos et al., 2012). The most recent great event was the Mw 8.8 Maule earthquake which struck southern Chile in 2010, generating a tsunami and causing 521 fatalities. However, large, shallow-crustal earthquakes are not uncommon in Chile. Since 1900 there have been 9 magnitude 7+ shallow-crustal ($<$15 km depth) earthquakes located in-land and therefore not directly associated with slip on the subduction megathrust (USGS, 2019). But most of these faults accumulate strain at slower rates compared to the subducting plate boundary and thus rupture infrequently.

The San Ramón Fault is one such fault. It runs along the foothills of the San Ramón mountains and bounds the eastern margin of the capital city Santiago, a conurbation which hosts 40% of the country's population within the city's Metropolitan region (7 million, 2017 estimates). Due to the rapid expansion of the city in the 20th and 21st century (Ramón, 1992), parts of the fault now lie beneath the eastern communes (districts) of the city (Fig. 1), in particular Puento Alto, La Florida, Peñalolén, La Reina and Las Condes. Yet it was only as recent as the past decade that Armijo et al. (2010) recognised that the San Ramón Fault is a Quaternary active thrust fault and poses a significant hazard to the city. Using field mapping and satellite imagery they estimated a slip rate of ∼0.5 mm yr$^{-1}$ for the fault, a much slower loading rate compared to the overall 7.5 cm yr$^{-1}$ plate convergence rate on the subduction zone. Palaeoseismic trench studies across the San Ramón Fault scarp revealed records of two historical ∼5 m slip events – approximately equivalent to a pair of Mw 7.5 earthquakes - at 17-19 k.yr and ∼8 k.yr ago (Vargas et al., 2014). However, based on geophysical investigations of the fault region Estay et al. (2016) concluded that the San Ramón Fault is segmented into four subfaults that are most likely activated independently in earthquakes with moment magnitude in the range 6.2 and 6.7. While Estay et al. (2016) do not discount the possibility of a larger rupture linking across all four segments, evidence from the trench studies (Vargas et al., 2014) suggest that larger magnitude earthquakes are possible on the fault. Therefore for the seismic hazard and risk analysis in this paper we take the worst case scenario of a complete rupture along the fault as our scenario case study.

Riesner et al. (2017) used balanced kinematic reconstructions of the geology across the region to deduce a long-term average shortening rate between 0.3 and 0.5 mm yr$^{-1}$, which is compatible with the earlier estimates by Armijo et al. (2010) and the recurrence-slip rates deduced from the paleo-earthquakes in the trench study by Vargas et al. (2014). This suggests that most of the active deformation across the West Andean Fold and Thrust Belt is accommodated on the San Ramón Fault. Therefore, despite the very low slip rates, the long time interval since the last earthquake means that significant strain has now accumulated on the fault and if it were to rupture completely, it could produce equivalent magnitude earthquakes as those recorded in the trench.

Pérez et al. (2014) performed a detailed analysis of local seismicity for the region and showed that the microseismicity at depth ($\sim$10 km) can be associated with the San Ramón Fault, implying that the fault is indeed active and accumulating strain. Vaziri et al. (2012) used Risk Management Solution's (RMS) commercial catastrophe risk modelling framework to estimate the losses from future earthquakes on the San Ramón Fault for Santiago. They estimate that a Mw 6.8 earthquake on the fault could result in 14,000 fatalities with a building loss ratio of 6.5%. However, the spatial distribution of these losses remains unclear.

Building on this previous work of identifying the hazard and losses, we aim to contrast the risk posed by the San Ramón Fault and place it in the context of other potential earthquake sources and a previous far-field subduction earthquake (Maule 2010). As we are examining the losses due to a very near-field source with exposed elements immediately adjacent to the potential rupture, we seek to delineate the location, extent and segmentation of the San Ramón Fault to improve the accuracy of the ground motion. Stereo satellite optical imagery is often used to derive high resolution digital elevation models (DEMs) over relatively large areas, which can be useful in identifying subtle active tectonic geomorphic markers of faulting as well as examine fault segmentation (Elliott et al., 2016). In this paper we use DEMs created from high resolution satellite imagery from the SPOT and Pléiades satellites (1.5 m and 0.5 m resolution respectively) to better characterise the surface expression of the San Ramón Fault and to also look for potential other fault splays within the city limits. Following a similar method to Chaulagain et al. (2016) and Villar-Vega and Silva (2017), and using the Global Earthquake Model's (GEM) OpenQuake-engine (Silva et al., 2014), we explore the contrasting losses to the residential building stock in the capital through scenario calculations for: a) future earthquakes on the San Ramón Fault, b) earthquakes on a hypothesised shallow splay buried beneath the centre of the city, c) deep intra-slab events, and d) the 2010 Mw 8.8 Maule earthquake. Our models help us to identify particularly vulnerable parts of the city and enable us to make targeted geographical recommendations to improve the seismic resilience of these communities.

We also explore losses to the non-residential building stock using the Risk Management Solution (RMS) commercial risk model. The RMS model provides a different view of the commercial risk, where the model is well calibrated due to the availability of losses from previous events, and covers the insured assets which is often one of the main mechanisms to recover from disasters.

## 2   Fault geomorphology from satellite imagery

Freely available global elevation data from the SRTM mission (Farr et al., 2007) has a spatial resolution of 30 m, which is insufficient to accurately map the San Ramón Fault scarp or look for other potential fault splays expressed in the geomorphology. To overcome the low resolution issue, we analysed SPOT6 stereo satellite imagery over a 35×36 km region covering Santiago city and the San Ramón mountains. The SPOT6 panchromatic imagery (acquired in 2014) has a spatial resolution of 1.5 m. We also tasked the acquisition of very high resolution (0.5 m panchromatic, acquired in 2016) Pléiades tri-stereo imagery over a smaller region (5×36 km) covering just the San Ramón Fault (Fig. 2). We used photogrammetry analysis using commercial software (ERDAS IMAGINE 2015) to produce topographic point clouds from the SPOT and Pléiades stereo imagery. We removed excessive low noise from the point clouds by initially doing a ground classification with only the highest points in a 3.5 m by 3.5 m grid and removing points that were highly isolated in wide and flat neighbourhoods, before redoing the ground classification with the filtered points. We then created raster gridded digital elevation models with 10 m and 2 m ground resolutions with the de-noised SPOT and Pléiades point clouds (∼83 million and ∼60 million points respectively). We did this by first triangulating the point cloud into a temporary triangular irregular network (TIN), and then rasterising the TIN into a digitial elevation model.

The San Ramón Fault is not immediately obvious in the Pléiades elevation map (Fig. 3), beyond the overall morphology of the uplifted San Ramón Mountains with a relief of 2.5 km above the Santiago basin. However the fault scarp is clear in the hillshaded DEM, the slope and Terrain Ruggedness Index (TRI) maps (Fig. 3a,iii-v) as a north-south trending lineament. The Terrain Ruggedness Index is a measure of the local variation in elevation about a central pixel (Riley et al., 1999; Wilson et al., 2007). A TRI of 0 indicates flat terrain while a value of 1 indicates extremely rugged terrain. Such analysis can highlight the change in elevation and slope at a fault scarp. We use these datasets to map the surface expression of the San Ramón Fault, confirming and building on previous work by Armijo et al. (2010) who used a 10 m DEM.

The identification of the active fault trace at the surface provides some evidence for the length, location and segmentation of the fault at depth, and that information is used as a constraint in the subsequent risk analysis for the range of earthquake sources that we seek to test. Measuring the vertical offset across the scarp can give an idea of the past activity along the fault, with the caveat that due to natural erosion processes, scarps tend to degrade with time. To do this we plotted a series of west-east profiles across the foothills of the San Ramón mountain to identify and measure the scarp height along the fault by determining the vertical offset between the best fit lines through the point cloud either side of the fault scarp (green and red lines in Figs. 4 and 5). The variable topographic slope along the fault means it is difficult to fit lines of equal length for each profile. In most cases we have tried to ensure a fit through at least 500m, but where possible 1km, of points either side of the fault scarp.

In the northern section we found scarp heights to vary between ∼5 m and ∼119 m along the the fault trace (Fig. 4). Profile *d* shows no clear evidence of a fault scarp, but since this profile is near a stream channel the scarp is moderated by fluvial erosion. However, it contains a clear break in slope, which is indicative of active faulting. Profiles *c* and *k* cross anticlines (∼145 m and ∼44 m high respectively) that have likely grown as a result of long-term movements in the hanging wall of the fault. The

anticline shown in profile *k* cuts across the northern section of an alluvial fan implying its growth post-dates the age of the fan deposit.

In the southern section the scarp heights vary between ∼2 m and ∼30 m (Fig. 5). Profiles *l* and *m* show two folds with heights of ∼23 m and ∼102 m respectively. The growth of the fold in profile *p* (∼68 m high) shows evidence that it blocked and diverted the Maipo river further south to its current position.

Our estimate of ∼33 m for profile *j* and ∼39 m for profile *g* are equivalent to the 31 m and 40 m estimated by Armijo et al. (2010). However, our estimate of ∼36 m for profile *h* is significantly less than their 54-60 m - the range is probably due to our interpretation of the upper slope which varies due to scarp degradation. This may be because of the relatively lower resolution DEM used by Armijo et al. (2010), 10 m, compared to our 2 m DEM.

Our geomorphic analysis of the surface trace of the San Ramón Fault confirms the findings of Armijo et al. (2010) for the central and northern part of the fault (red line in Figure 3), and extends the trace of the fault further to the south (blue line in Fig. 3).

In their trench study Vargas et al. (2014) measured fault displacements of the order of ∼5 m. Projected to the vertical for a 45 degree dipping fault this gives a vertical offset of about 3.5 m. Given that our average measured scarp height is ∼32 m, it is likely that this represents cummulative displacement over numerous earthquakes. However there is significant variation along the fault with the smallest scarp of 2 m and the largest at 119 m. This variation is probably due to erosion along the mountain front leading to variable degradation of the fault scarps. The smaller scarps represent the cummulative displacements of fewer earthquakes. While these observations enable us to determine the active fault segments that comprise the San Ramón Fault, the variations in scarp height mean it is difficult to trace specific historical ruptures along the fault.

Our observations of the fault traces show a network of fault segments that are ∼0.5 km to ∼8 km long. Our assumption is that at depth these segments represent the same fault. This is motivated by field observation that large earthquake ruptures often consist of multiple rupture segments (Barka et al. (e.g. 2002); Civico et al. (e.g. 2018)). It is possible that the two main strands of the San Ramón Fault (Figs. 4 and 5) could rupture independently. However, these individual fault segments are not long enough to produce a mangitude 7.5 earthquake from a 5 m slip event, which justifies exploring earthquake scenarios that rupture across both strands of the fault.

The West Andean frontal faults drawn by Armijo et al. (2010) appear to terminate at the northern and southern margins of the city. Although it is possible that the San Ramón Fault accommodates the full shortening across the region, it is also possible that the frontal faults extend further west beneath the city (Fig. 2b), hidden by the sediments of the central depression. Our investigations using the SPOT satellite DEM and point cloud data do not show any clear evidence of a fault scarp within the central regions of the city. However, this could be masked by urban development or the fault could be buried, as in the case of the Pardisan thrust fault beneath the city of Tehran in Iran where no primary fault is visible at the surface (Talebian et al., 2016). Similarly, the 2011 Mw 6.3 Christchurch (New Zealand) earthquake occurred on a previously unrecognised fault buried right beneath the centre of the city (Elliott et al., 2012), but its impact was much greater than the larger earthquake (Mw 7.1) that struck the year before, but outside of the city.

Riesner et al. (2017) proposed a first-order model for the deeper structure of the San Ramón Fault and found that it constitutes the frontal expression of a major west vergent fold-and-thrust belt that extends laterally for thousands of kilometres along the western flank of the Andes (see also Armijo et al., 2015). Since it is well known that frontal faults of fold-and-thrust belts tend to migrate out of the central highlands through progressive growth of new faulting (e.g. Davis et al., 1983; Dahlen, 1990; Reynolds et al., 2015), it is not unreasonable to assume that younger faults would extend further west from the San Ramón Fault. We project the location of this inferred fault along strike from the West Andean cordillera frontal fault (Fig. 2b) and assume the dip is the same as for the San Ramón Fault. In section 3 we will explore the losses from moderate magnitude earthquakes (Mw 6 and Mw 6.5) on this hypothesised buried fault within the city with larger magnitude events on the San Ramón Fault (Mw 7 and Mw 7.5), consistent with the palaeoseismic trench work of Vargas et al. (2014). The magnitudes for the central Santiago splay scenario were determined using standard fault scaling relationships (Wells and Coppersmith, 1994), where a rupture with length ∼25 km, width ∼12 km and a coseismic slip of 1 m would result in an earthquake with moment magnitude in the range 6-6.5.

In 1647 a large earthquake destroyed Santiago, which at the time was a 100 year old Spanish town, and killed an estimated one-fifth of its inhabitants (de Ballore, 1913; Udías et al., 2012). Details of this earthquake remain poorly understood, and there is much debate on the epicentral location (e.g. Lomnitz, 1983; Comte et al., 1986; Lomnitz, 2004). Lomnitz (1970) notes that historical descriptions of the damage indicate an epicentre within 50 miles of Santiago at most, while Poirier (2006) mentions that the earthquake did not produce any devastating tsunamis; both of which point to a source on a fault near the city. As there is no evidence of this earthquake in the trench studies along the San Ramón Fault (Vargas et al., 2014), it is unlikely that the earthquake originated there as suggested by Rauld (2002). While it is possible that the earthquake occurred on one of the faults in the principal cordillera (Farías et al. (2010)) our assumption is that the the main activity in region is on the frontal portions of the fold-thrust-belt ((e.g. Dahlen, 1990; Reynolds et al., 2015)). It is also possible that the 1647 earthquake occurred on the West Andean Faults to the north and south of the city (Fig. 2), however, due to the lack of any evidence for either case we feel it is reasonable to explore the worst case scenario of an earthquake ocurring on the extension of these faults through the city.

Another possibile candidate for the 1647 earthquake is an earthquake in the subducting slab beneath the city. Therefore, we also examine intra-slab faulting scenarios. This is motivated by the most damaging earthquake in terms of fatalities in south central Chile in the previous century. The 1939 earthquake (Ms ∼7.8) caused ∼28,000 deaths (many times more than the Great 1960 subduction earthquake) and produced extensive damage to the city of Chillan (Saita, 1940; Frohlich, 2006), about 200 km south of Santiago. Beck et al. (1998) modelled the P-wave first motions for this earthquake and concluded that it was a normal faulting event within the down-going slab at a depth of 80-100 km. Since the subducting slab beneath Santiago is also about 80 km beneath the city (Hayes et al., 2012), we explore the losses from similar normal-faulting events in the slab beneath Santiago.

## 3 Earthquake scenarios for the residential building stock

The development and implementation of measures to minimise the physical impact due to earthquakes requires a comprehensive understanding of the potential for human and economic losses, which is usually achieved through earthquake risk assessment studies (e.g. Silva et al., 2015b; Chaulagain et al., 2016). For risk management purposes, risk is the potential economic, social and environmental consequences of hazardous events that may occur in a specified period of time (see Grossi and Kunreuther (2005) for details).

We use the GEM OpenQuake-engine v3.3.2 (Silva et al., 2014; GEM, 2019) to calculate the damage and losses to residential buildings from earthquake scenarios on predetermined faults for all 52 communes that make up the Santiago Metropolitan Region (∼1.1 million buildings). In the sections below we briefly describe the key components of the damage and risk calculation: the exposure, the hazard and vulnerability (Fig. 6).

### 3.1 The residential building exposure model

In order to describe the residential building stock of the Santiago Metropolitan region we used the exposure model established by Santa-María et al. (2017). The exposure model was built using data from the national population and housing census surveys (2002 and 2012), and information from the 2002–2014 Formulario *Único de Estadísticas de Edificación* (Unique Edification Statistic Form, UESF). The exposure model describes the number and distribution of residential buildings at the census block resolution, and contains information on the main material of construction, number of storeys (Fig. S2), age of construction, expected ductility, the number of people living in each building, and the replacement cost per unit area. The replacement cost includes an estimate of the structural, non-structural and content costs of each building. We assume the earthquake scenarios occur at night and therefore the residential fatality estimates represent the night time losses. Table 1 summarises the most important information in the residential building exposure model. The most commonly used building material for residential buildings is masonry (79% of all buildings) with confined masonry the dominant building typology (39% of total buildings), followed by reinforced masonry (26%) and unreinforced masonry structures (14%). To improve computing efficiency we resample the Santa-María et al. (2017) exposure model from the census-block resolution to a 1×1 km grid (Fig. S1). Our exposure model reveals that Puento Alto, Maipú and La Florida are the most populated communes, together accounting for about 26% of all residential homes in the Santiago Metropolitan Region. Puento Alto and La Florida are centred on the San Ramón Fault.

Figure 7 shows the fraction of the total building stock in each commune categorised into the 5 building classes against the percentage of people living below the poverty line - defined as USD400 per month for a family of 4 (Ministerio de Desarrollo Social, 2016). The coefficient of determination ($R^2$) for the fit through each building class is 0.63, 0.01, 0.45, 0.20, 0.73 for reinforced concrete (RC), confined masonry (MCF), reinforced masonry (MR), unreinforced masonry (MUR) and wooden (W) buildings respectively. It is clear that the scatter in the data for the masonry buildings is large and reflected in the low $R^2$ values, particularly for MCF buildings implying little correlation between levels of poverty and the fraction of confined masonry buildings. However, we find that the fraction of reinforced concrete (RC) buildings decreases significantly with the

proportion of people living below the poverty line. This trend is balanced by an increase in the fraction of wooden (W), reinforced masonry (MR) and unreinforced masonry (MUR) buildings with level of poverty

## 3.2 Definition of the earthquake scenarios

Unlike probabilistic seismic hazard analysis where the risk calculation is initiated with a stochastic event dataset, in this study we calculate the damage and losses for specific earthquake scenarios on pre-determined faults. We chose a scenario based approach because it provides a clear communication of the relative scale of potential damage and losses from the recently recognised proximal San Ramón Fault versus that from the better characterised offshore subduction faulting, which is important for emergency management planning and for raising societal awareness of risk (Silva et al., 2014).

We modelled the San Ramón Fault as a set of four rectangular slip planes (total length of 35 km) to account for the changes in geometry along strike due to fault segmentation from our DEM analysis (Fig. S3). The prescribed fault planes dip at 45 degrees to the east and extend from the surface down to 12 km depth, based on the structural cross sections drawn by Armijo et al. (2010) and the depth of microseismicity determined by Pérez et al. (2014) as indicators of the down-dip width that is locked and accumulating strain. The location of the hypothesised splay fault is in line with the West Andean Front, and 12 km west of the San Ramón Fault, consistent with the approximate 10 km spacing inferred in the major thrust faults beneath the San Ramón-Farellones Plateau (Pérez et al., 2014). We represent the splay fault using a single, 45 degree eastward dipping rectangular plane extending from 0.5 km below the surface down to 12 km depth with a north-south strike, running 25 km along longitude 70.65°W. The deep intra-slab fault scenario is modelled using a single, westward dipping rectangular plane of length 35 km in the subducting slab beneath the city. We used a 70 degree dip for the intra-slab fault to represent a similar earthquake to the 1939 Chillan earthquake for which Beck et al. (1998) estimated a 60–80 degree dipping fault plane. We used the Slab 1.0 model (Hayes et al., 2012) to set the top depth and bottom depth of the fault plane at 85 km and 98 km respectively for this locality. Earthquakes on the San Ramón and Santiago splay faults are prescribed as having a pure thrust mechanism (rake +90°) while the intra-slab are normal (rake −90°). The fault characteristics are summarised in Table S1.

The hazard component of the calculation concerns determining the spatial pattern of the key shaking parameters from each scenario event by employing a Ground Motion Prediction Equation (GMPE). The hazard parameters used here are Peak Ground Acceleration (PGA) and the Spectral Acceleration (SA). There are many GMPEs available in the literature (see Douglas and Edwards (2016) for a review and www.gmpe.org.uk for an updated compendium). For our analysis we use three equations for shallow crustal earthquakes (Akkar et al., 2014; Bindi et al., 2014; Boore et al., 2014), and two for the intra-slab scenarios calculations (Abrahamson et al., 2016; Montalva et al., 2017). These were selected for the OpenQuake-engine by expert opinion during the Global GMPEs Project (Stewart et al., 2012, 2015), and updated since. Averaging several selected GMPEs helps to partially propagate the epistemic uncertainty of the distribution of shaking that arises from a non-perfect knowledge of ground motion. There are several methods to calculate the distance from each exposure point to the rupture. To remain consistent across the GMPEs we implement the form of the equations that use the Joyner-Boore distance, defined as the shortest horizontal distance from each exposure element to the surface projection of the rupture area. However, we present the damage and loss results at the district level by calculating the sum of the losses of all points within each district.

For each scenario we produce 1000 realisations of the ground motion in the region to account for the aleatory variability in the ground motion, and assume the entire fault ruptures in the earthquake. We account for the spatial correlation of the intra-event variability during the generation of each ground-motion field, to ensure assets located close to each other will have similar ground-motion levels, according to the methods described by Jayaram and Baker (2009).

For the 2010 Mw 8.8 Maule earthquake, we directly used the USGS ShakeMap as the input ground shaking for the damage and risk assessment calculations (see Villar-Vega and Silva (2017) for details of this procedure).

## 3.3   Site effects

The Santiago metropolitan region is located in a narrow basin between the Andes and coastal mountains filled with quaternary fluvial and alluvial sediments (Armijo et al., 2010). Using numerical simulations of the Santiago basin taking account of the superficial geology, Pilz et al. (2011) showed that there is a strong and sometimes complex basin amplification effect on the peak ground velocity from hypothetical earthquakes on the San Ramón Fault. While in this study we are unable to account for the complexities of basin resonance and topography, we attempt to take into account the basin amplification effect in our ground motion calculations by using the Vs30 velocities - the shear wave velocity in the top 30 m of soil.

In this study two datasets were used to obtain the Vs30 information for Santiago. The first consists of local microzonation studies, which contain seismic zonation maps (Pasten, 2007; Leyton et al., 2011), and proposed Vs30 values for soil types in each zone taking into account additional information from soil penetration tests (Humire-Guarachi, 2013). However, given the cost and time demand of such studies, microzonation maps are usually focused on limited areas. Therefore, for the remaining zones we supplemented the micronzonation data using velocities from the USGS Global Vs30 Map Server (Allen and Wald, 2007). This method derives maps of seismic site conditions using topographic slope as a proxy, assuming that stiffer materials (i.e. higher Vs30 values) are more likely to maintain a steep slope, while deep basin sediments are deposited mainly in environments characterised by a lower velocity.

Figure 8 shows the Vs30 values used in this study at the building exposure locations, with the values from the microzonation studies indicated in circles. Note that this will probably be an underestimate of the full basin effects (Joyner, 2000). For example, not accommodating for basin resonance will mean that our models do not take into account the particular vulnerability of buildings of certain height prone to resonance, which was an important factor for example in the Kathmandu rupture and basin amplification in Nepal with resonance at 4-5 seconds (Galetzka et al., 2015).

## 3.4   Building fragility and vulnerability models

The physical, or structural, vulnerability for a built system is defined as its susceptibility to suffer losses when subjected to earthquake shaking. In our scenario calculations we use two main forms of vulnerability models: fragility functions, which are used to relate earthquake shaking to certain levels of physical damage to a building (e.g. extensive damage, collapse), and vulnerability functions (structural and occupants), which relate the earthquake shaking to a structure with the economic and human losses.

Villar-Vega et al. (2017a) analytically derived fragility functions for the 57 building classes in the exposure dataset developed for the South America Risk Assessment (SARA) project (Yepes-Estrada et al., 2017). For our analysis we use the subset of these equations that represent the building exposure in the Santiago Metropolitan region (Table 1 & Fig. S1). To derive the fragility functions, Villar-Vega et al. (2017a) represented the structural capacity of each building class by a set of single-degree-of-freedom (SDOF) oscillators. Each oscillator was subjected to a suite of ground motion records representative of the South American tectonic environment and seismicity using GEM's Risk Modelers Toolkit (Silva et al., 2015). From each analysis, the maximum spectral displacement of each SDOF was used to allocate it into a damage state (e.g. collapse). In this paper, we focus our scenario analysis on the spatial distribution of collapsed buildings, which comprises not only the physically collapsed buildings but also partially collapsed structures (Villar-Vega et al., 2017a).

The simplification of each building typology to a single-degree-of-freedom oscillator means that the calculated fragility functions only approximate the building response to ground shaking. Therefore these would not be sufficient to investigate builing-by-building scale losses from earthquake shaking. However, we believe it is sufficient to explore aggregated district level losses. And so, while the scenario calculations are done one a 1×1 km gridded exposure model, the losses presented in the following sections aggregate these to the district level.

A vulnerability curve establishes the probability distribution of a loss ratio (e.g. fatalities/total number of occupants), given a shaking intensity measure level (Fig. 9 and Fig. S5). Vulnerability curves are generally empirically derived using loss data, usually collected through insurance claims or governmental reports. A database of fragility and vulnerability functions can be found in the OpenQuake platform (Yepes-Estrada et al., 2016; Martins and Silva, 2018). We used these vulnerability functions to directly model fatalities and repair costs, where the loss ratio for the former would be the ratio of fatalities to exposed population, and for the latter the ratio would be that of repair cost to cost of replacement for a given building typology.

## 3.5 Residential building collapse and loss results

The median predicted ground motion for the larger earthquake scenario considered for each fault is given in Fig. 9. They show the relatively simple ground motion patterns from the single rectangular Santiago splay fault and a more complex pattern from the San Ramón Fault. For the San Ramón case most of the high ground shaking is around the communes to the east of the city, while for the splay fault case there is a more even distribution of shaking across the central communes.

Our damage and loss results, averaged over the GMPEs used in each scenario calculation, reveal that the collapsed building estimates for each scenario are distributed unevenly across the city (Fig. 10). Figure 11 shows a summary of the damage and loss results for all scenarios. It is clear that the damage and losses are greater for the larger magnitude earthquake considered in each case - as one would expect since larger earthquakes, at a given depth, produce higher intensity ground shaking.

For the San Ramón scenarios the losses are mostly concentrated in the communes around the fault. Most collapsed buildings are located in Puento Alto (23,100–28,800; 16–20%), La Florida (12,400–15,700; 15–19%) and Las Condes (10,500–12,800; 20–25%) where the first numbers in the brackets are the building collapse counts for the two San Ramón Fault scenarios and the second two numbers the percentage collapse of the total number of exposed buildings in the commune. We calculate fewer residential building collapses in Peñalolén (5,900–7,400; 14–17%) and La Reina (5,000–6,200; 18–23%) despite these

communes also being located on the fault. This discrepancy could be explained through a combination of greater exposed population, therefore more residential buildings (Fig. S6 shows the percentage of collapsed buildings), and the level of poverty.

Puento Alto has the largest population of these communes (622,356) and also the greatest percentage living below the poverty line (Table 2). Puento Alto and La Florida also generally contain a greater proportion of masonry constructions - 93% and 86%, compared to 79%, 71% for Peñalolén and La Reina respectively - which perform poorly in the San Ramón earthquake scenarios. While Peñalolén has a low fraction of RC residential buildings (5%), which generally perform well in our calculation, it is compensated by a large proportion of wooden structures (17%) which perform the best when subject to seismic shaking.

In general the greatest percentage of collapsed buildings (Fig. S6) occur, as expected, in the communes directly on the fault - e.g. Vitacura (3,000–3,600; 21–25%) and Las Condes (10,500–12,800; 20–25%). However there are several communes with high collapse fractions that are not located on the fault, notably Ñuñoa (8,400–10,600; 20–25%) and Macul (4,200–5,300; 18–23%). These both have a high fraction of unreinforced masonry buildings, 27% and 13% respectively compared to an average of 9% for the communes on the fault (Table 2). Unreinforced masonry buildings are the most likely building class to collapse in all the scenarios considered in this study (Table 4).

In terms of anticipated fatalities for the larger San Ramón scenario (Fig. 12), the communes of Ñuñoa and Providencia (10 km west of the San Ramón fault trace) are modelled as experiencing the highest fatality rates of 4–5 per thousand (Fig. S7). In terms of absolute numbers, the largest number of fatalities (Fig. 12) occur in Ñuñoa and Las Condes (1,120–1,420, 1,080–1,330 respectively). Overall the fatalities across the region are estimated in the range 9,700–12,700 (Fig. 11), and a fatality rate of 0.15–0.19% (Table S3). The residential losses in terms of replacement costs average 8–10 billion USD (5–7% mean loss ratio). The greatest replacement costs are for Santiago (1.3 billion USD), but are also high (0.5+ billion USD) for the communes of Puento Alto, Las Condes and La Florida on top of the San Ramón fault, as well as for Ñuñoa further west (Fig. S8).

For the earthquake scenarios on a buried fault splay beneath the centre of the city, the distribution of collapsed residential buildings is similar to the San Ramón scenario with damage concentrated towards the eastern communes of the city in the hanging wall of the fault. Most collapses occur in Puento Alto (13,600–20,900; 9–15%) and Santiago (10,000–15,400; 17–27%). As in the case of the San Ramón Fault the high collapse count in Puento Alto probably reflects the large number of residential buildings in that commune. Of the communes directly next to the fault splay, Santiago has the largest number of residential buildings (57,341). However, the greatest impact in terms of collapse fraction is in the communes in the central districts near the fault, with the highest fraction collapse occurring in Santiago (10,000–15,400; 17–27%), Providencia (3,400–5,500; 15–25%), Independencia (2,500–3,700; 16–24%) and Ñuñoa (6,600–10,300; 15–24%). The estimated fatalities for the buried splay scenarios are similar to slightly above those for the San Ramón cases despite being a magnitude less in scale, in the range 6,500–11,500 (0.10–0.17% loss ratio). The most affected commune are also similar, including Santiago, San Miguel, Providencia and Ñuñoa with fatality fractions of 4–5 per thousand for the larger Mw 6.5 scenario (Fig. S7). The greatest number of fatalities for both magnitudes (Fig. 12) are also in Santiago and Ñuñoa (870–1,600 and 710–1,310 respectively,

Table S3). The residential replacement costs are 6.1–9.6 billion USD (4-6% loss ratio) for the two magnitude scenarios (Table S3). The greatest losses are in Santiago, Ñuñoa and Puento Alto (Fig. S8).

The overall collapse count for the magnitude 7 deep intra-slab scenario is small, but the magnitude 7.5 scenario results in a substantial number of collapsed buildings (about 60,000), with most collapsed homes and fatalities (Fig. 12) generally located in the more populous communes. The extent of collapse across the city is more diffusive due to the buried nature of the intra-slab source, with building collapse up to 8% in the centre (Lo Espejo, San Joaquin and Independencia). The estimated total number of fatalities (Fig. **??**) for the larger event is 3,180 (0.05%), with the largest number of fatalities (200–300, Table S3) each in Santiago, Ñuñoa, Maipú and Puento Alto (Fig. 12). The estimated replacement cost is 3 billion USD, 2% loss ratio) with the greatest costs distributed in the same commune as fatalities (Fig. S8).

Central government statistics estimate a total collapse count of 81,444 residential buildings throughout Chile in the 2010 Mw 8.8 earthquake, with most damage occurring in the Maule, Biobío, O'Higgins and Santiago Metropolitan regions (Elnashai et al., 2010; de la Llera et al., 2017) with 4,306 of these occuring in the Santiago Metropolitan region (yellow star in Fig. 11). While the collapse count is smaller than our modelled estimate of $9,800 \pm 8,000$, it is within the error margin. The discrepancy could have arisen due to a slightly different exposure model. The actual exposure in 2010 would have been different than our exposure model estimates, which uses data from 2014. Moreover, there is often ambiguity regarding the classification of actual structural collapse and damage beyond repair (and thus in need of demolition). See Villar-Vega et al. (2017a) for a discussion on this topic.

While we estimate building collapse fractions up to 21% (Providencia) for the Mw 7 San Ramón scenario, the average collapse ratio across all the communes is $\sim$8%. This is larger than the 6.5% estimated by Vaziri et al. (2012) for a magnitude 6.8 earthquake on the fault. However, their estimate of 14,000 fatalities is larger than the 9,700 we estimate for a Mw 7 earthquake. This difference is most likely due to variations in the exposure model and calculation procedure (i.e. choice of ground motion models). But since Vaziri et al. (2012) used an industry exposure model we are not able to determine the exact cause behind the difference in our estimates.

In general across all the scenarios considered in this study, the largest number of collapsed buildings occur in the highly populous communes (with therefore more buildings) close to the fault. However, the collapse and fatality fraction, number of collapse/fatalities over the exposure (Fig. S6 and S7), reveal particularly vulnerable areas. Several communes experience relatively large damage and loss fractions, which is an indication of the vulnerability of the communities in these communes. Of particular importance are Ñuñoa, Providencia and Santiago which generally have high loss fractions with 3, 3 and 2 fatalities per thousand and 15%, 15%, 15% damage fraction respectively averaged over the 6 earthquake scenarios. In comparison the average loss fraction across all communes and all scenarios is 0.8 fatalities per thousand and 7% building collapse. Therefore targeted measures to retrofit particularly vulnerable residential buildings (unreinforced masonry) could reduce the seismic risk faced by communities living in these communes.

## 4 Non-residential insured losses

We also used Risk Management Solutions' (RMS) commercial Chile Earthquake Model, developed in 2011, with the most recent Industry Exposure Database (IED) to derive industry loss estimates for specific earthquake scenarios. The exposure model contains only non-residential building information and does not include public infrastructure such as roads or bridges. The exposed values (or 'total insured value', TIV) in this dataset includes commercial buildings, contents, and business interruption, and are aggregated at the comuna level.

Table 3 and Fig. 13a gives a summary of the average Gross loss ratios - calculated loss, over the total insured value - for the maximum magnitude scenarios on the San Ramón and Santiago splay faults. The Gross losses are the full replacement costs to the property after accounting for insurance penetration and after the application of deductibles, limits and co-insurance. This is often referred to as the insured loss. It is worth noting that these losses are a subset of the full economic losses in an earthquake, since the Gross losses account for insurance penetration, which is always less than 100%. The average insurance penetration (residential and non-residential) in the Santiago Metropolitan region was 31.5% in 2011. However, only an estimated 30% of small commercial business owners had earthquake insurance, which rises to greater than 75% for large commercial and industrial facilities (Muir-Wood, 2011).

## 5 Discussion

Whilst a detailed past record of earthquakes and variability of recurrence on the San Ramón Fault is not precisely known, the palaeoseismic work of Vargas et al. (2014) tentatively points towards a recurrence interval of the order of ∼8 k.years; determined from records of two past earthquakes at 17-19 k.years and ∼8 k.years ago. Given that the last event was ∼8 k.years ago it is prudent to consider a San Ramón rupture scenario of Mw 7.5 as a real possibility to plan for. Vargas et al. (2018) find that present urbanisation of eastern Santiago reached 55% of the San Ramon fault trace, evidencing that this active geological structure has not been considered in urban regulations developed for the metropolitan region.

### 5.1 Residential collapse by building class

Buildings of differing construction material type are known to perform markedly differently under seismic shaking (Park and Hamza, 2016). We aim to identify the expected better and less well performing building classes exposed in Santiago under our varying earthquake shaking scenarios. In order to compare the collapse ratios for the different building classes defined in Table 1 (RC, MCF, MR, MUR, W) we calculate the normalised collapse fraction, $NCF_t^s$, for each building class according to:

$$NCF_t^s = \frac{(c_t^s/e_t)}{(C^s/E)} \tag{1}$$

where $e_t$ is the number of exposed residential buildings of building class $t$, and $c_t^s$ is the number of collapsed buildings of the same class in earthquake scenario $s$. $C^s$ is the total number of collapsed buildings in scenario $s$ and $E$ is the total number of exposed buildings in the city.

Therefore, if $NCF_t > 1$, then typology $t$ is more likely to collapse than the average. The results of this normalisation is shown in Table 4. Across the 6 earthquake scenarios, we find that reinforced concrete (RC) residential buildings perform best with a normalised collapse fraction, $NCF_{RC} = 0.5$. It is also clear that unreinforced masonry (MUR) structures collapse the most across all scenarios with an average $NCF_{MUR} = 2.6$, implying that MUR structures are over 2.5 times more likely to collapse than the average. Typically, 3–13% of buildings in the most affected communes (Table 2) are unreinforced masonry construction except for Santiago and Macul where 27% of the buildings are MUR. This is why we typically observe relatively large collapse fractions in Santiago (1,700–15,400; 3–27%) and to a lesser extent Macul (600–5,300; 3–23%) over the 6 earthquake scenarios considered in this study. Wooden residential homes (W) perform very well with an average $NCF_W = 0.2$. Confined masonry ($NCF_{MCF} = 0.9$) and reinforced masonry ($NCF_{MR} = 0.8$) perform better than unreinforced masonry buildings and slightly better than the average. It is clear that masonry construction in general performs worse during earthquakes.

## 5.2   Magnitude-distance trade-off

The calculated collapse and losses for each of the scenarios using the residential building exposure show the expected pattern of greater losses for a larger earthquake on a given fault (Fig. 11 and Table 5). Our calculations show that the San Ramón earthquake scenarios are the most damaging to the city with 136,400 residential building collapses from a magnitude 7 and 181,300 collapses from a magnitude 7.5 earthquake on the fault, a difference of 25%. The estimated damage from the buried Santiago splay fault are 107,300 collapses for a magnitude 6 and 172,200 collapses for a magntiude 6.5, a difference of 38%. We note a similar pattern in the losses with an increase of 24% and 20% in fatalities and replacement cost respectively between a magnitude 7 and 7.5 earthquake on the San Ramón Fault, and an increase of 38% and 43% on the splay fault earthquakes. It is clear from our calculations that a half magnitude increase in earthquake size results in more damages and losses from the santiago splay fault than the San Ramón Fault. This is probably because shaking from the splay fault exposes more of the densely populated communes in the centre of the city where a small increase in shaking can have more impact.

In all cases, the magnitude 8.8 Maule earthquake produces fewer losses to the city than the smaller local earthquakes, despite having 100–10,000 times the moment release of the other scenarios considered here. According to our models, it is clear that there is a trade-off between earthquake magnitude and distance in terms of residential building collapse and fatalities. Chatelain et al. (1999) also found a similar trade-off in their earthquake risk assessment for Quito, Ecudaor. Therefore, simply focusing on large offshore megathrust earthquakes would mask the significant risks posed from moderate size earthquakes on smaller but more local active faults.

## 5.3   Residential and non-residential insured losses

Fig. 13b shows the loss distribution of the residential building replacement costs due to a magnitude 7.5 earthquake on the San Ramón Fault and a magnitude 6.5 earthquake on a buried splay fault beneath the centre of the city. While these are not directly

comparable with the non-residential insured losses (Fig. 13a), partly due to the fact that the insured losses are impacted by insurance penetration and also includes business interruption and policy conditions, there are some clear differences between the two that are worthy of note.

For the San Ramón scenario the highest residential building replacement costs are generally concentrated in the communes with large populations (hence more residential buildings) close to the fault, and those with a large fraction of reinforced concrete buildings, which are more expensive than masonry structures. The greatest losses occur in Puento Alto, La Florida, Las Condes and Santiago; while the insured losses had equally high losses in all communes along the fault, including high loss ratios in Lo Barnechea and Huechuraba, two communes not directly above the fault.

This difference reflects the concentration of high value commercial properties in the more affluent eastern communes, where businesses are more likely to be insured (Muir-Wood, 2011), particularly in Las Condes and La Reina. The residential losses reflect the damages in highly populous communes, as evidenced by the losses in Santiago, Ñuñoa and La Florida.

Similarly the losses in residential buildings for a magnitude 6.5 earthquake on the Santiago splay are concentrated in the eastern communes in the hanging wall of the fault (Santiago, Ñuñoa and Puento Alto), while the insured losses concentrate around the central business districts of Santiago and Huechuraba.

## 5.4 Caveats and limitations

There are a number of caveats/sources of uncertainty in estimating damage and losses from past and hypothetical earthquake scenarios using the method described in this paper. One of the main sources of uncertainty is the exposure model, which does not exactly correspond to the exposed population and building portfolio that was affected by a past earthquake. For this study the exposure model was developed by Santa-María et al. (2017), using information from census surveys and housing information from 2014. Therefore, it is important to note that the exposure model used in investigating the damage and loses for the hypothetical earthquake scenarios on the San Ramón, Santiago splay and deep intra-slab faults does not represent the exact current exposure of the Santiago Metropolitan Region. This is particularly important for Santiago where rapid eastward expansion of the city into the foothills of the San Ramón mountains puts an increasingly greater population closer to the San Ramón Fault (Figure 1) and onto its hanging wall, where ground accelerations are typically higher. Therefore, methods that allow for a near-continuous update of building inventories and locations are needed to maintain the veracity of exposure.

One of the largest sources of uncertainty in the calculations are in the GMPEs. In order to capture the epistemic uncertainty in both median ground-motion predictions and their associated aleatory variability, we used several equally weighted GMPEs (e.g. Bommer et al., 2005, 2010). The differences in the datasets used to derive each GMPE and the way each GMPE calculates the ground motions is partly why the uncertainty range on our estimates of the number of collapsed buildings are large (Fig. 11).

The development of fragility models also involves large uncertainties. The fragility functions used in this study were developed using a probabilistic approach where a set of structures are tested against a suite of ground motion records (Villar-Vega et al., 2017a). Since it is time and cost prohibitive to develop a fragility function for every building in the city, each of the buildings are allocated to a building typology within a general building class in the exposure model. Each typology is then

represented by a single-degree-of-freedom block to calculate its response to the ground motion records. It is important to note that this level of simplification adds uncertainty in the final response of any particular typology in the event of an earthquake (see Villar-Vega and Silva (2017) for a discussion). Also, it is not possible to say how any individual building will respond to an earthquake using this approach, as the fragility functions do not include the unique complexities in design and construction of every building.

The results shown in this paper are from scenario based calculations on predetermined faults, and thus we cannot provide the relative likelihood of a shaking event as in a Probabilistic Seismic Hazard (PSHA) framework. However, this is an important start to motivate continued work on the the recurrence interval of faults in the region, begun by Armijo et al. (2010) and Vargas et al. (2014) on the San Ramón fault. The SPOT DEM is not good enough to conclusively identify the presence of any geomorphic marker that may result from a splay fault within the city. This remains an open question as to its very existence, let alone relative level of activity. Higher resolution DEMs or detailed field surveys might be able to resolve this issue.

It is important to note that although the focus of this paper has been to explore the direct damage and losses due to earthquakes, in the case of an actual event there are often cascading hazards in the form of liquefaction, tsunamis, landslides and blocked waterways leading to floods, fires etc. that can lead to loss of lives and livelihoods (Gill and Malamud, 2014). Nevertheless, historically most fatalities in earthquakes were due to direct building collapse - apart from the large tsunami death tolls from a few 21st century megathrust earthquakes (Ambraseys and Bilham, 2011).

## 6  Conclusions

In this study we use high resolution DEMs of the metropolitan area of Santiago city and the foothills of the San Ramón mountains, created using SPOT and Pléiades satellite imagery, to accurately map the surface expression and segmentation of the San Ramón Fault. We recognise that estimates of the impact from specific earthquakes (historical or hypothetical) can support decision makers in the development of risk reduction strategies. We therefore use the OpenQuake-engine to calculate damage and losses for realistic earthquake scenarios on the mapped San Ramón Fault as well as a potential unknown buried fault directly beneath the centre of the city, and a deep intra-slab fault. We compare these loses with those for the 2010 magnitude 8.8 offshore Maule earthquake as a reference level of a recent event. Our calculations show that there is a strong magnitude-distance trade-off in terms of direct damage to the exposed building portfolio and fatalities, with smaller more local shallow earthquakes causing greater losses to the city than a larger offshore megathrust earthquake. It is clear that the eastward expansion of the city into the foothills of the San Ramón mountains has exposed a large number of, predominantly affluent, people to a future earthquake on the San Ramón Fault. While the recurrence interval of large earthquakes on the fault are long ($\sim$8 k.years) and only very loosely constrained, the last Mw 7.5 event was $\sim$8 k.years ago (Vargas et al., 2014). So it is prudent to consider the potential impacts of a San Ramón rupture scenario of similar magnitude. We calculate losses using scenario based models of San Ramón earthquake ruptures in the magnitude range 7–7.5 under the current residential exposure. Our models estimate $181,000 \pm 80,000$ partial to total building collapses, $12,700 \pm 4,500$ fatalities and replacement costs of $10 \pm 3$ billion USD for the larger magnitude earthquakes the San Ramón Fault can accommodate, assuming all fault segments

fail at once. While these numbers are subject to considerable uncertainty arising from a changing exposure, uncertain ground motion prediction equations and fragility functions, they provide an informative guide to the potential scale of losses from a large earthquake on the San Ramón Fault. For all modelled scenarios, the most vulnerable building class is unreinforced masonry, while wooden structures and reinforced concrete are the most resilient to earthquake shaking. Therefore, effective near-term risk reduction measures could target unreinforced masonry homes for retrofitting campaigns, particularly in Ñuñoa, Providencia and Santiago, while in the mid-long term a drive towards reinforced concrete homes would significantly reduce the risks to future earthquakes both from near and far-field sources. This work also reinforces the need to identify active faults adjacent to or beneath cities in actively deforming zones, and the need to update the exposure models as such cities encroach onto these faults. We have highlighted that local crustal earthquakes in the magnitude range 6–7.5 can have a much greater impact than distant larger earthquakes. Therefore the frequency of distal major earthquakes has to be balanced by the potential for infrequent but much more potent local smaller earthquakes on less active faults.

*Code and data availability.* The latest version of the OpenQuake-engine can be downloaded from the Global Earthquake Model GitHub repository: https://github.com/gem/oq-engine. The point clouds created from both the SPOT and Pléiades stereo satellite imagery have been uploaded to the OpenTopography platform (http://opentopography.org) and are available to download for free.

*Competing interests.* Any opinions, findings and conclusions or recommendations expressed in this manuscript are those of the authors and do not necessarily reflect those of Risk Management Solutions, Inc.

*Acknowledgements.* This work has been supported by the NERC/AHRC/ESRC Global Challenges Research Fund (GCRF) awarded to the Seismic Cities project (grant number NE/P015964/1) and also by the Royal Society GCRF Challenge grant (CHG\R1\170038) and in part by the British Geological Survey (BGS) Overseas Development Aid program (NEE6214NOD). This paper is published with the approval of the BGS Executive Director. We would like to thank the Seismic Cities team for helpful discussions, especially Paula Repetto for help in accessing the Chilean datasets. The lead author would like to thank Catalina Yepes-Estrada, Anirudh Rao and Marco Pagani as well as all other members of the GEM Hazard and Risk team for their patience in explaining the details and methodologies of the OpenQuake-engine, and the Research Center for Integrated Disaster Risk Management (CIGIDEN) in Chile for hosting the January 2017 OpenQuake-engine training workshop. We also thank Tim Craig for suggesting the need to also consider intra-slab earthquakes as potential major shaking sources and Laura Gregory for help with interpreting fault profiles. We thank colleagues at RMS including Justin Moresco, Chesley Williams and Robert Muir-Wood. We gratefully acknowledge the CEOS Seismic Pilot for providing the Pléiades stereo imagery over the San Ramón Fault. ©CNES (2016), distribution Airbus DS/Spot Image. Pleiades images made available by CNES in the framework of the CEOS WG Disasters. JRE is supported by a Royal Society University Research fellowship (UF150282).

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

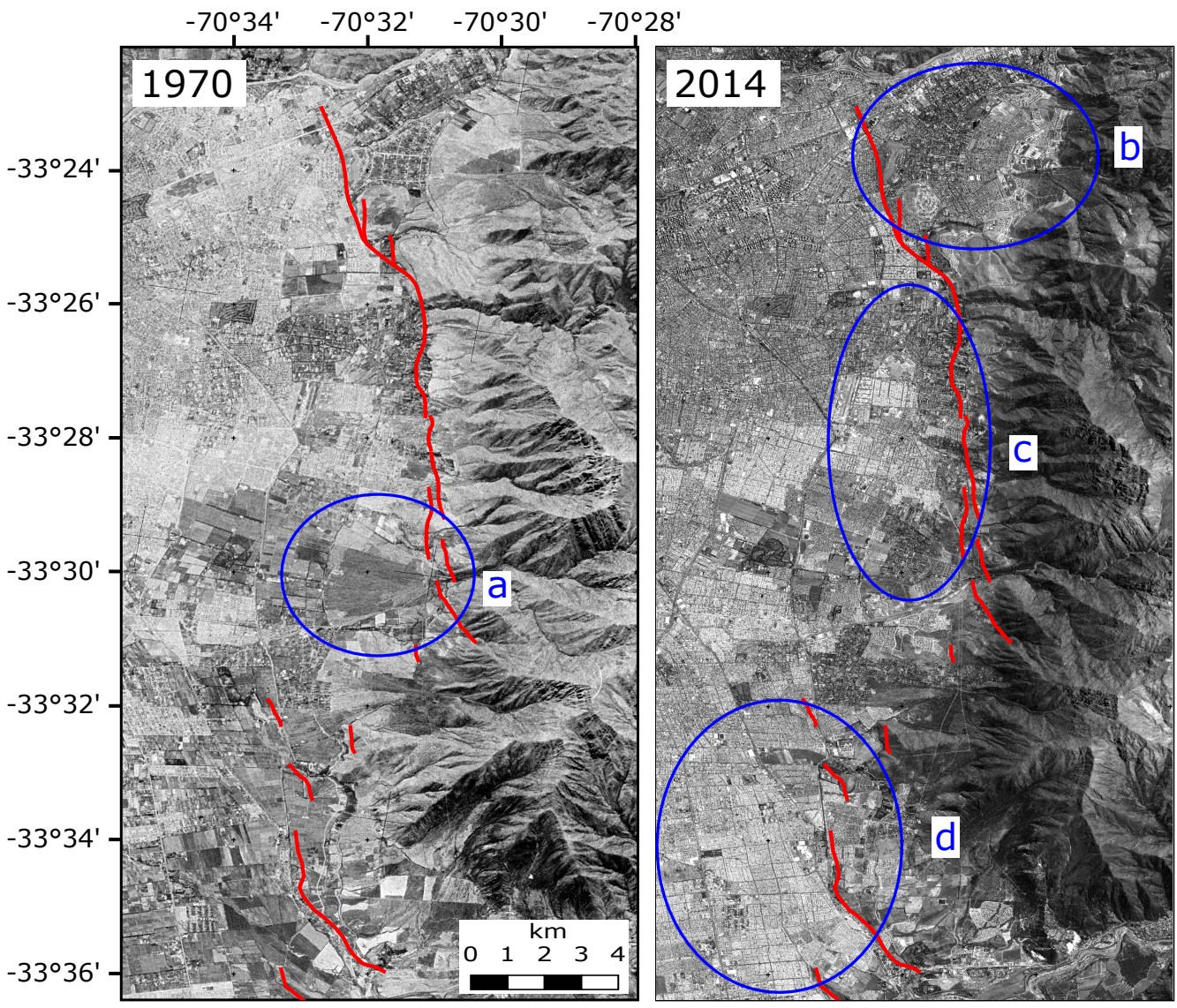

**Figure 1.** Declassified corona satellite image (∼2 m resolution) from 1970 (left) and the same region in the SPOT imagery (1.5 m resolution, right) used in this study showing the eastward expansion of the city over the San Ramón Fault (red lines). Four notable regions are highlighted in blue. a) An alluvial fan that is clearly visible in the older imagery but completely covered with buildings in the recent image. b) Expansion into the foothills of the mountain onto the hanging wall of the San Ramón thrust fault. These are often more affluent neighbourhoods with better views across the city. c) Urban densification in the central regions. d) Land use change from farmland to dense urban neighbourhood masking the fault trace.

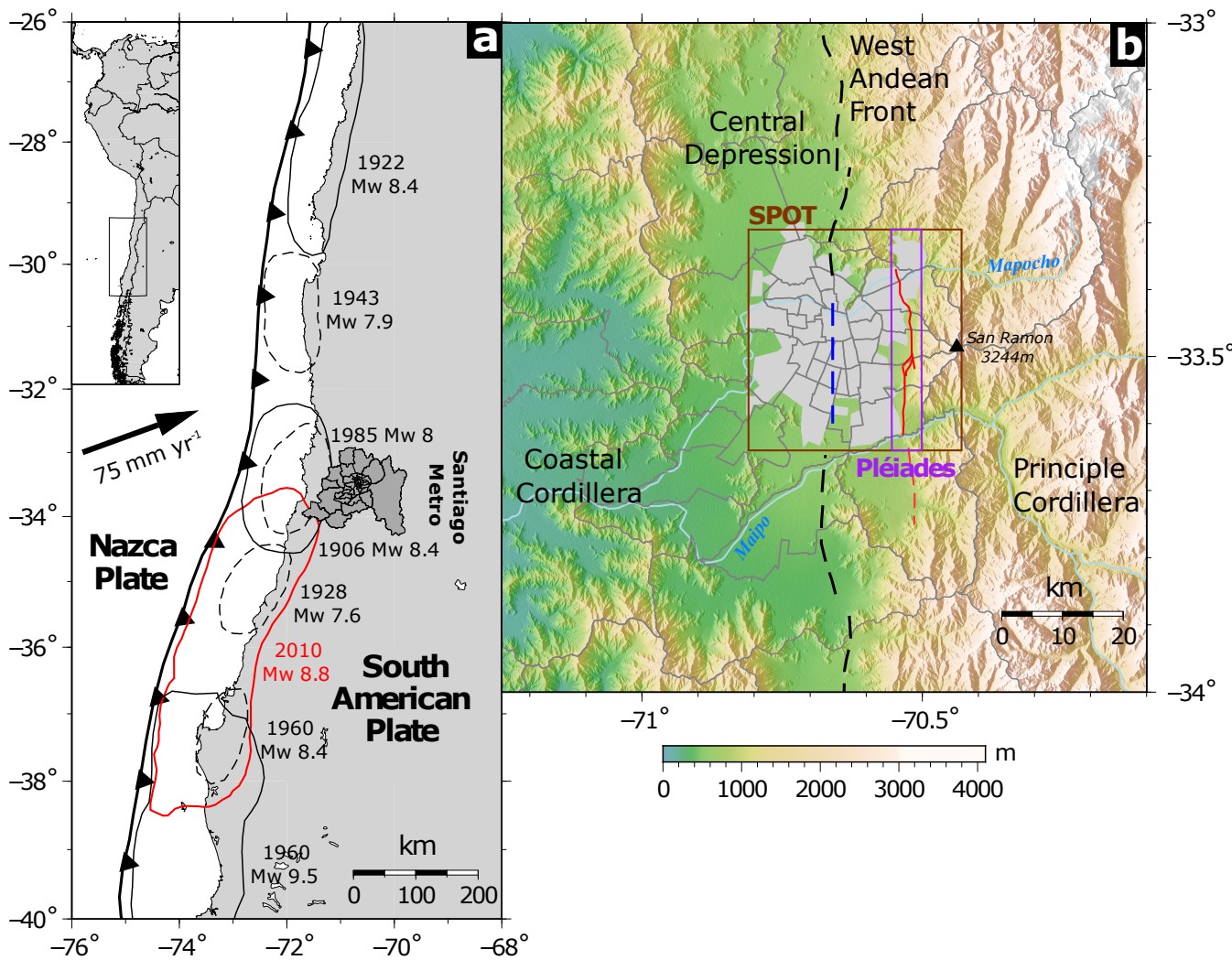

**Figure 2.** a) Map of central Chile showing the location of great earthquakes for the past century on the subduction zone where the Nazca plate is converging beneath the South American plate at a rate of 43 mm yr$^{-1}$ (Zheng et al., 2014). The Santiago Metropolitan region is shown in the dark grey outline, subdivided by commune (names of all the communes are given in Fig. S1). b) A 90 m SRTM shaded terrain map of the region around Santiago city (light grey). The SPOT and Pléiades satellite data used in this study cover the region shown by the maroon and purple polygons respectively. The San Ramón Fault is shown in red while the blue dotted line is the location of our inferred buried fault within the city (see text for details). The dashed black lines are the mountain front faults mapped by Armijo et al. (2010).

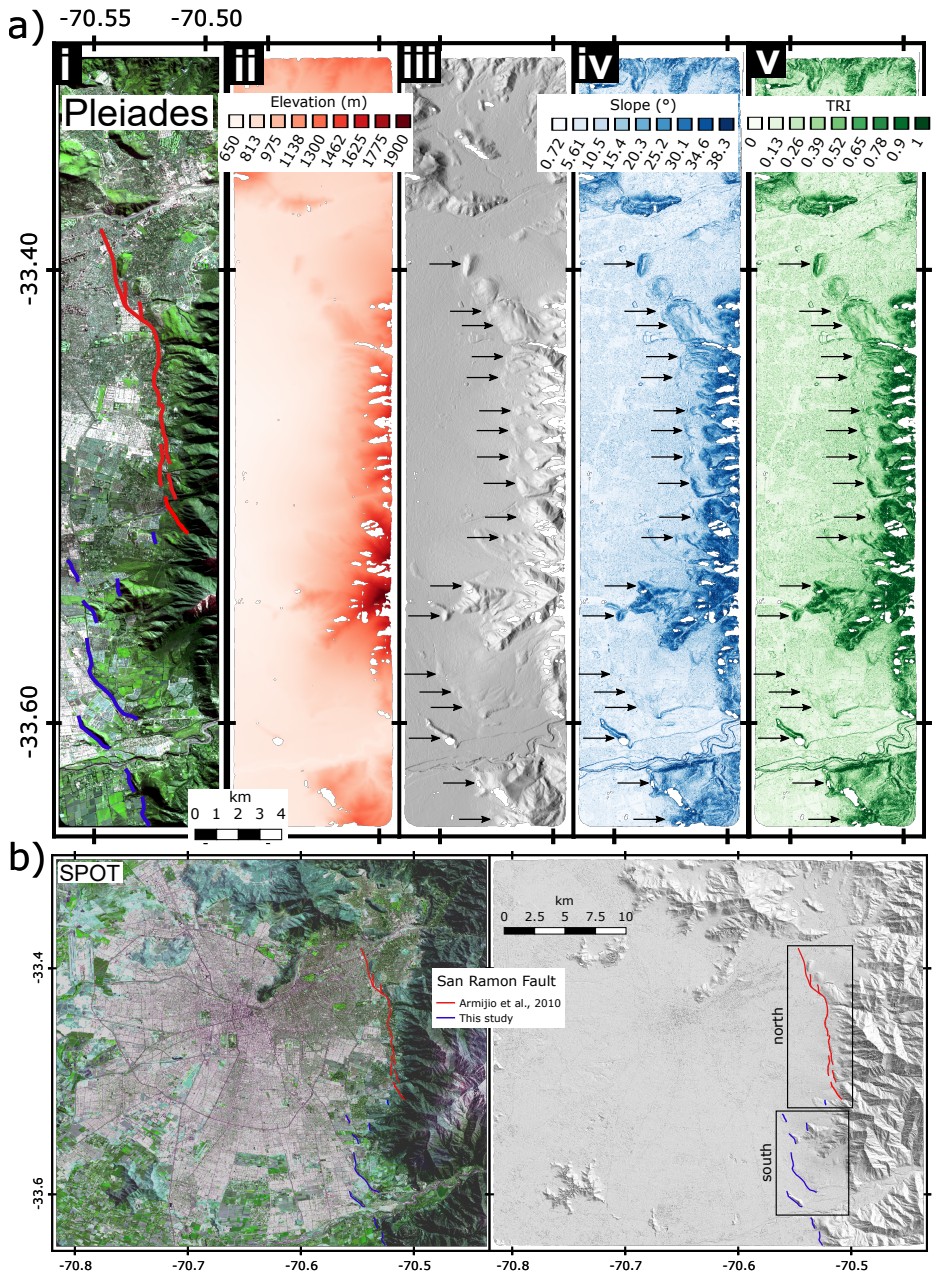

**Figure 3.** a) i - The Pléiades satellite optical multispectral image, ii - the elevation map created using photogrammetry analysis of the panchromatic optical image, iii - the hillshaded digital elevation model (DEM), iv - slope map and v - the terrain ruggedness index (TRI). Data gaps are on steep slopes in shadow resulting in low contrast and inability to derive heights from stereo image matching. b) The SPOT satellite multispectral image (left) and the resulting hillshaded DEM (right) derived from stereo panchromatic pairs.

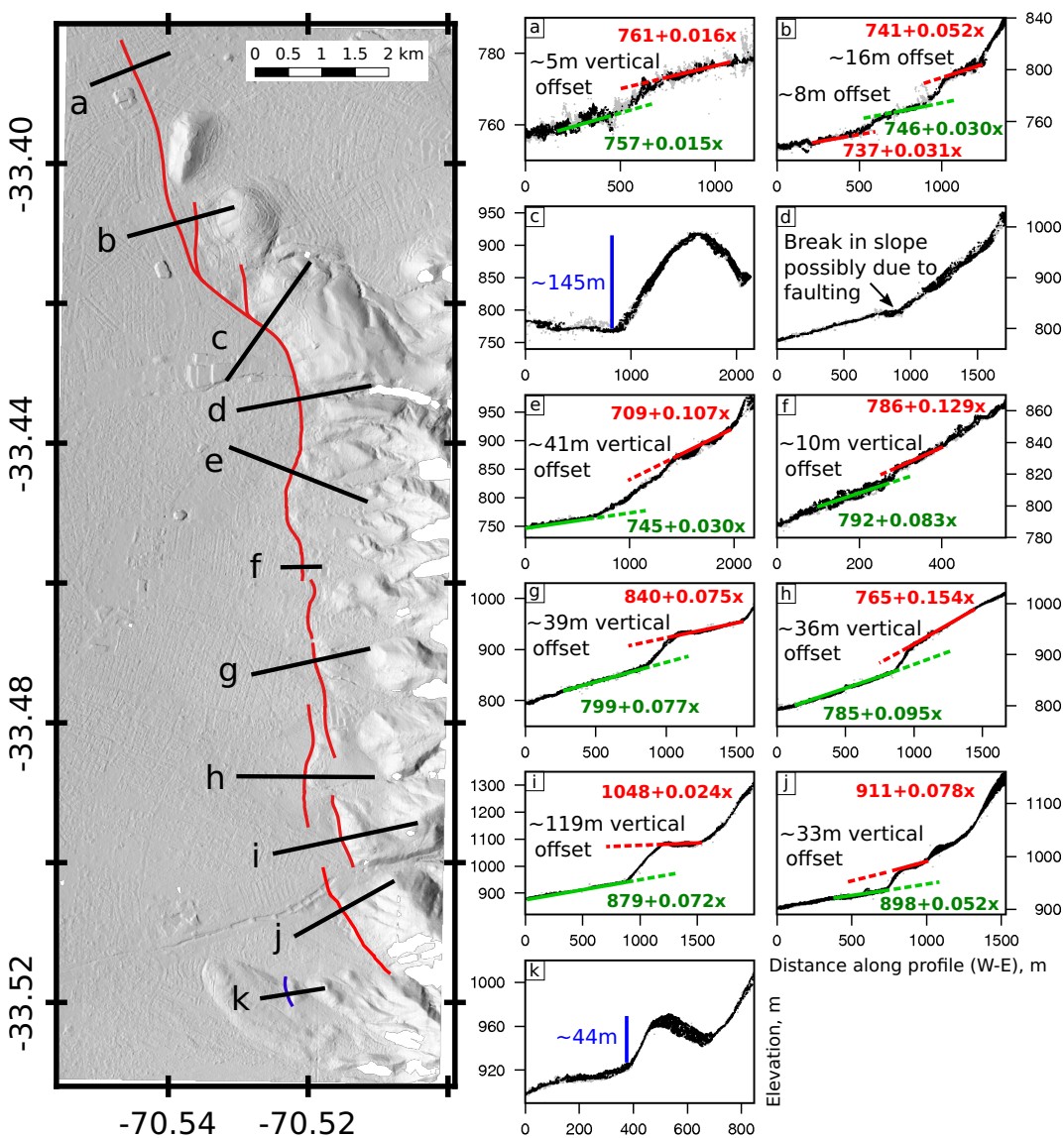

**Figure 4.** The northern section of the Pléiades-derived DEM (2 m resolution) indicated in Fig. 3. The black points on the profiles are ground pixels within a 30 m swath of the profile line from the Pléiades imagery-derived point cloud while the grey are from the SPOT point cloud. The red and green lines are best fit lines through the point clouds either side of the fault scarps. The scarp height is estimated from the vertical offset between these two lines. The blue lines are the height of anticlines measured from the downslope side.

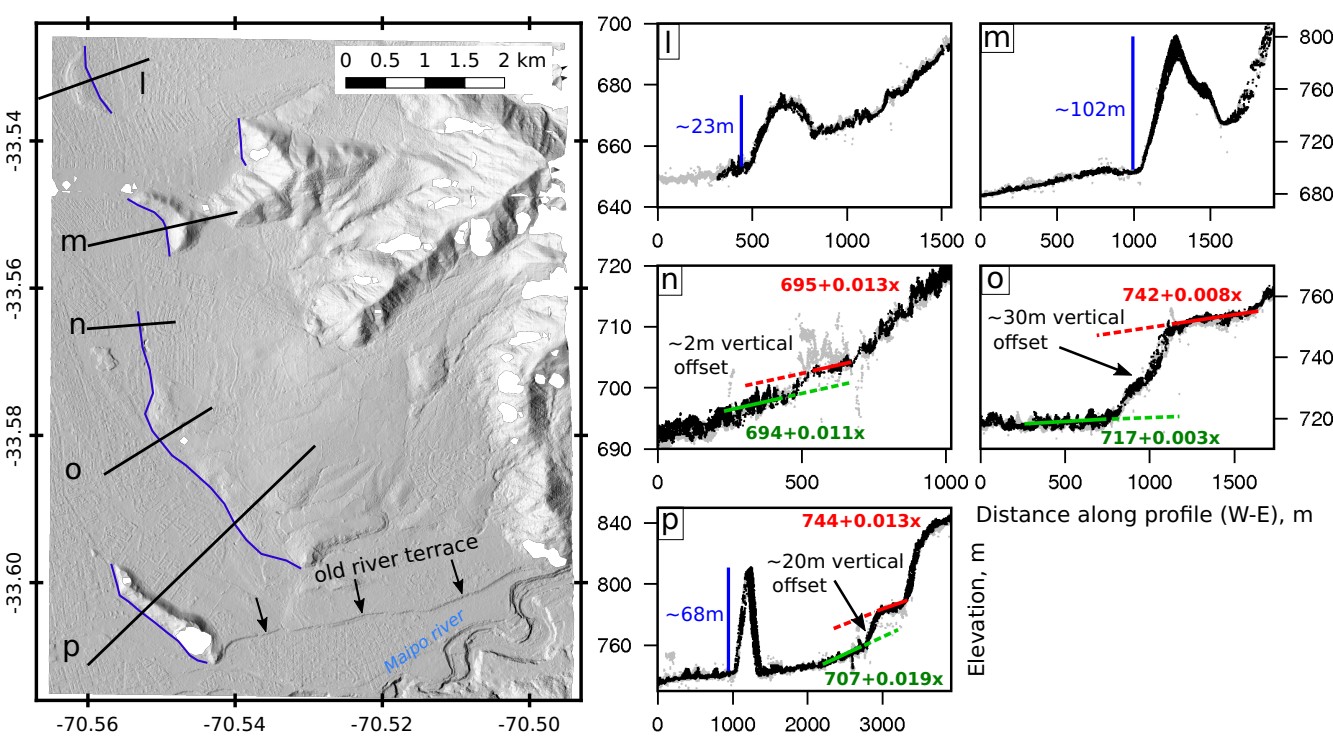

**Figure 5.** The southern section of the Pléiades DEM indicated in Fig. 3. The black points on the profiles are ground pixels from the Pléiades imagery-derived point cloud while the grey are from the SPOT point cloud.

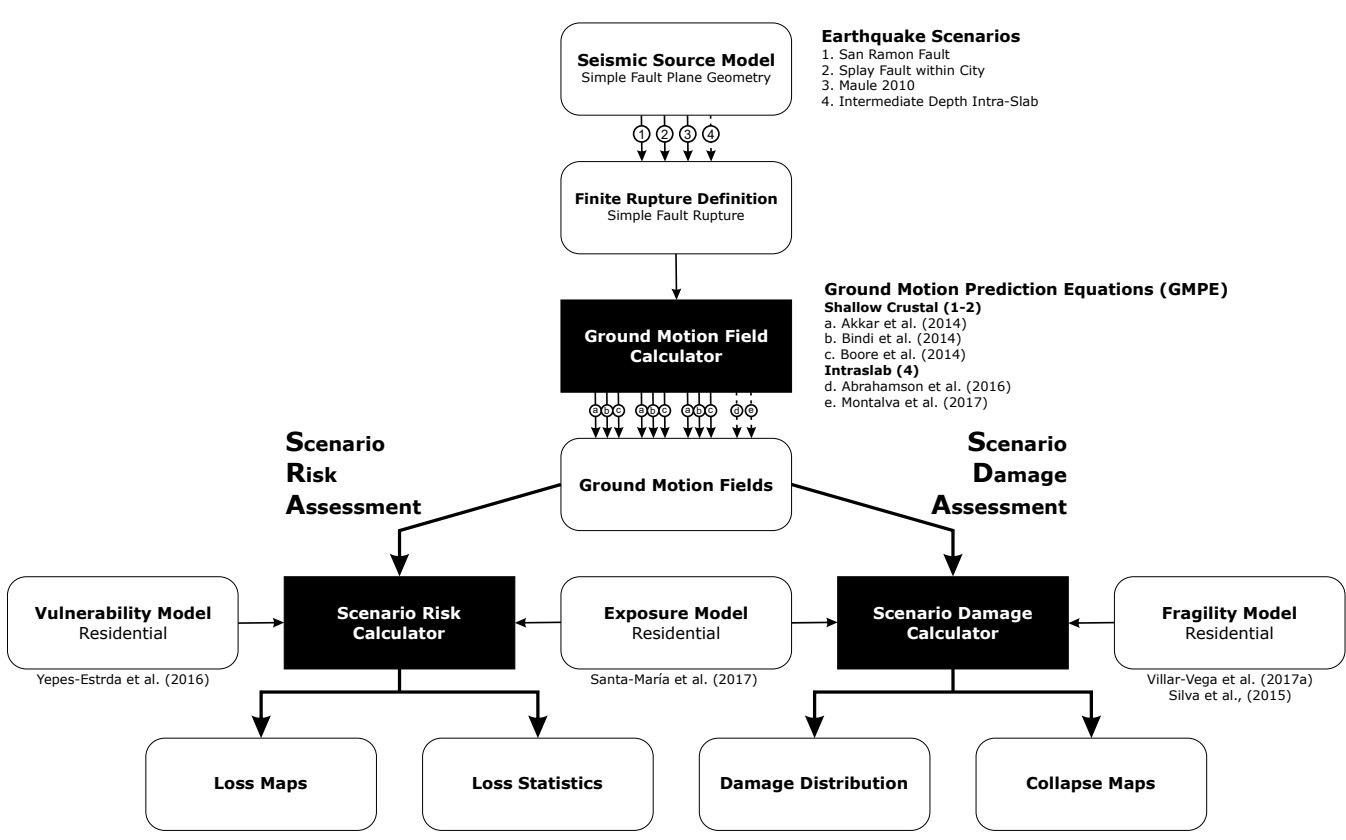

**Figure 6.** A graphical representation of the damage and loss calculation work flow using the Global earthquake Model's OpenQuake-engine (Silva et al., 2014). Black boxes represent model calculators while white boxes are data input/outputs.

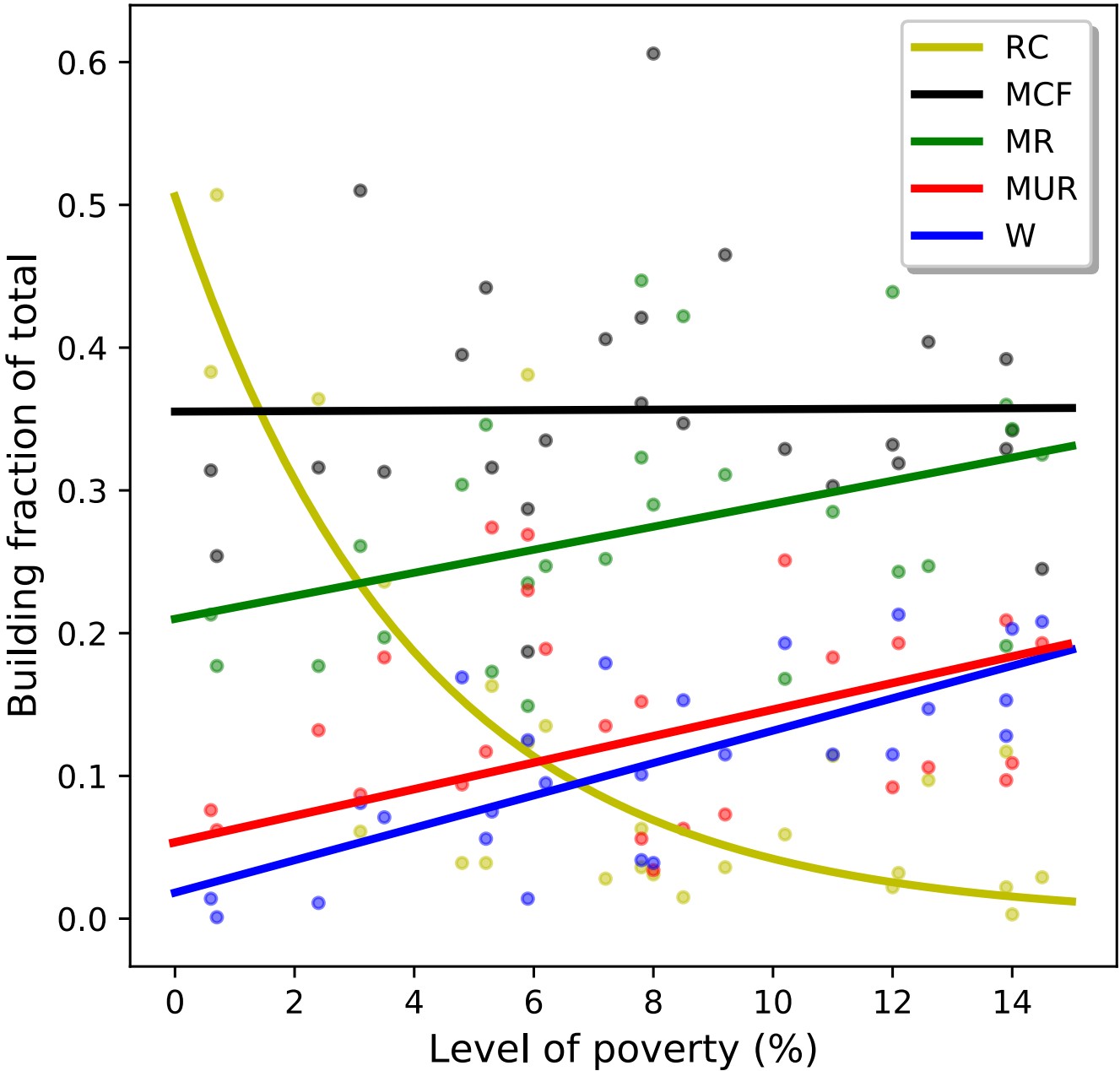

**Figure 7.** The fraction of residential buildings by building class - RC is reinforced concrete, MCF is confined masonry, MR is reinforced masonry, MUR is unreinforced masonry and W is wooden construction - against the proportion of people living below the poverty line in the communes of Santiago Metropolitan region. Solid lines represent best fit trends through the data (linear for all cases except exponential for reinforced concrete) with coefficient of determination of 0.63, 0.01, 0.45, 0.20, 0.73 for RC, MCF, MR, MUR and W respectively. The poverty line is defined as USD400 per month for a family of 4 (Ministerio de Desarrollo Social, 2016).

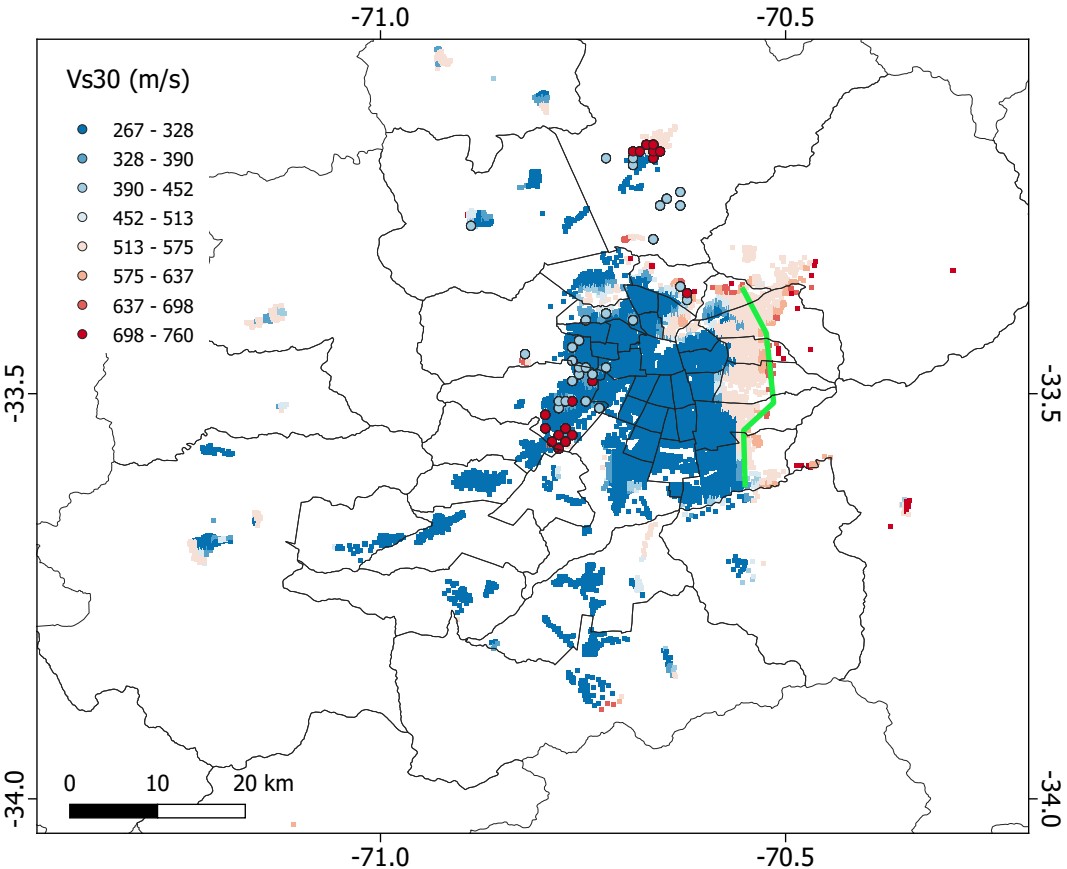

**Figure 8.** A map of the Vs30 - shear wave velocity in the top 30 m of soil in m s$^{-1}$ at the exposure locations. The circled data are from microzonation studies (Leyton et al., 2011; Humire-Guarachi, 2013) and the remaining are estimates derived from the topographic slope (Allen and Wald, 2007). The green lines indicate the surface trace of the San Ramón fault used in the seismic risk scenario calculations. Red colours indicate a relatively high Vs30 velocity and are generally in regions with exposed/shallow bedrock. Relatively slow Vs30 velocities are associated with sedimentary basins.

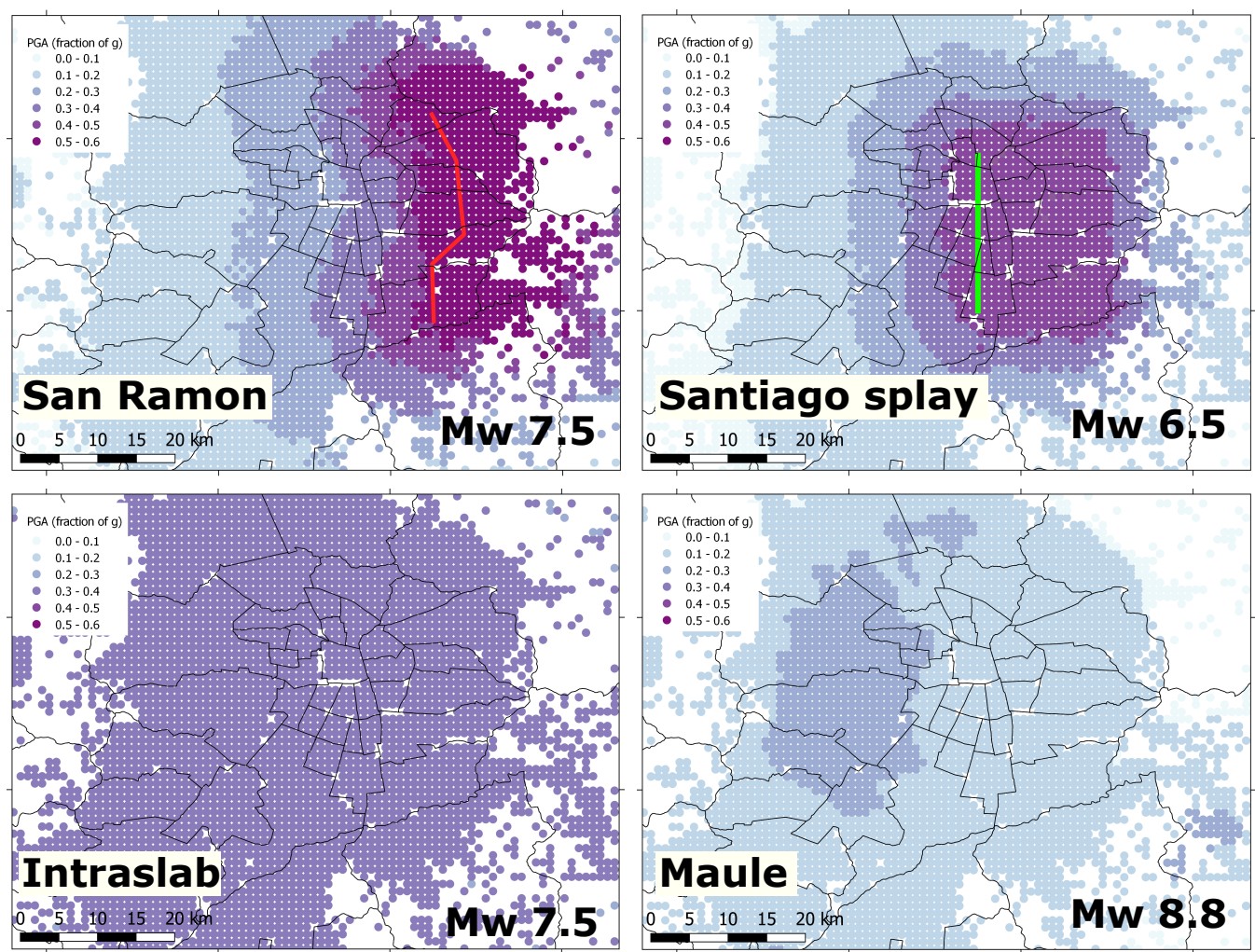

**Figure 9.** The estimated median Peak Ground Acceleration (PGA) as a fraction of *g* for the the largest earthquake scenario considered for each fault. For the San Ramón (red line) and Santiago splay fault (green line) cases, these represent estimates using the Akkar et al. (2014) ground motion prediction equation, while those for the intraslab fault are estimates from the Abrahamson et al. (2016) equation. The USGS peak ground accelerations for the Mw 8.8 Maule earthquake is shown at the bottom. The ground motions for the full set of earrthquake scenarios are given in Fig. S4.

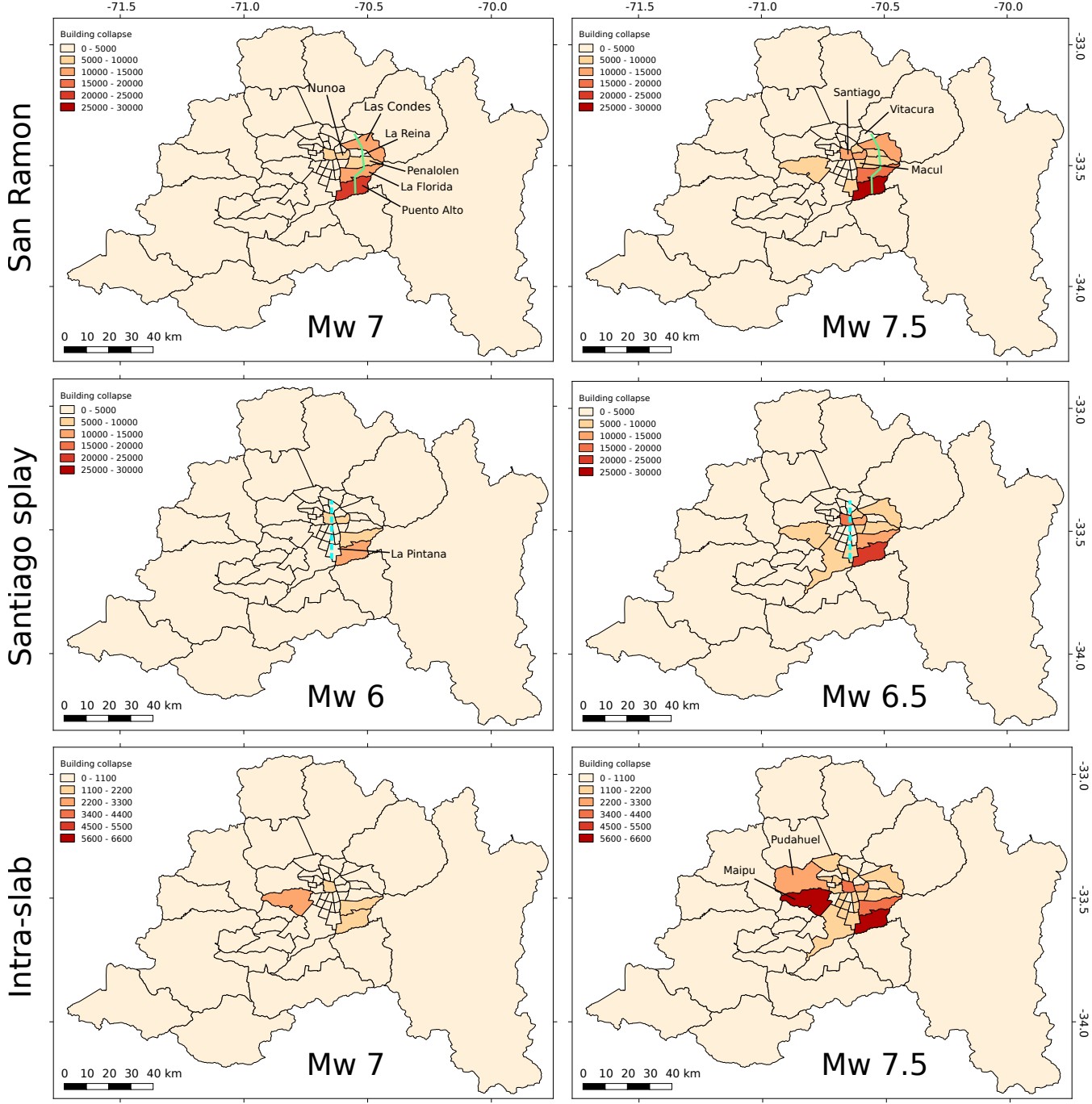

**Figure 10.** The distribution of collapsed buildings for the earthquakes considered in each set of magnitude pair scenarios for the San Ramón Fault (green line, top row), the Santiago splay fault (dashed cyan line, middle row) and a deep intra-slab fault (bottom row). The collapse counts are the average for the GMPEs used in each calculation and include both the complete collapse and partial collapse total count. Note that the range of the colour scale changes between different upper four and lower two panels. The collapse fraction for each commune are given in the supplementary material (Fig. S6). Names of all the communes are given in Fig. S1

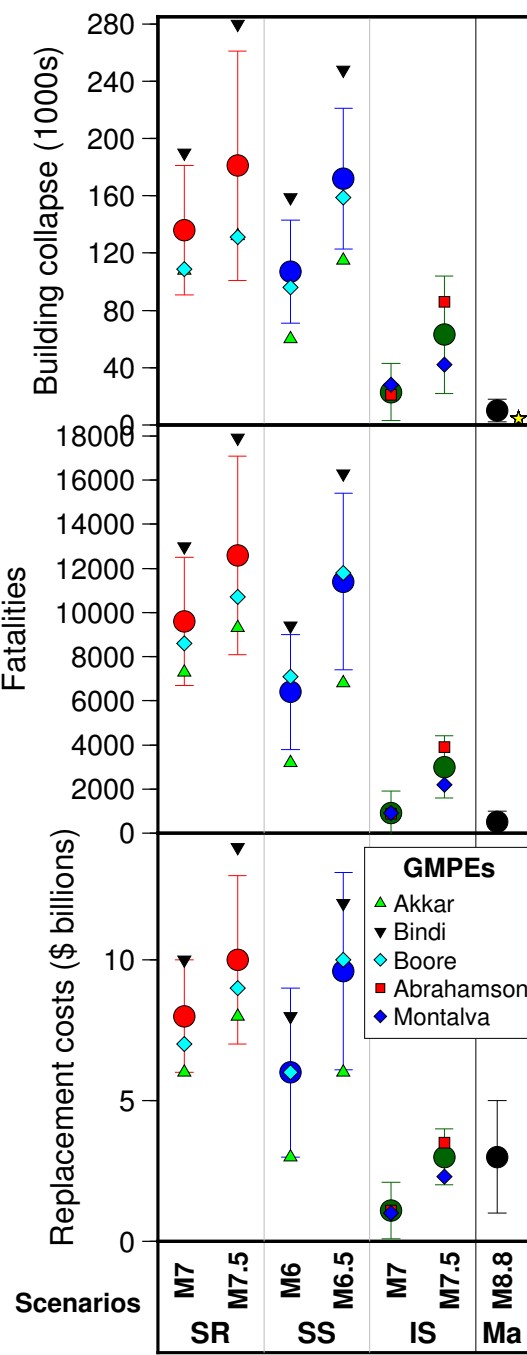

**Figure 11.** A summary of the total number of building collapse, fatalities and replacement costs for each scenario calculation. The solid circles are the average across the GMPEs considered in each scenario, which are indicated by the smaller polygons. The spread in the estimates from the GMPEs indicate the epistemic uncertainty in our calculations. Error bars represent 1 standard deviation determined from the 1000 Monte Carlo simulations. The yellow star denotes the actual number of building collapses (4,306) in Santiago in the 2010 Maule earthquake (Elnashai et al., 2010).

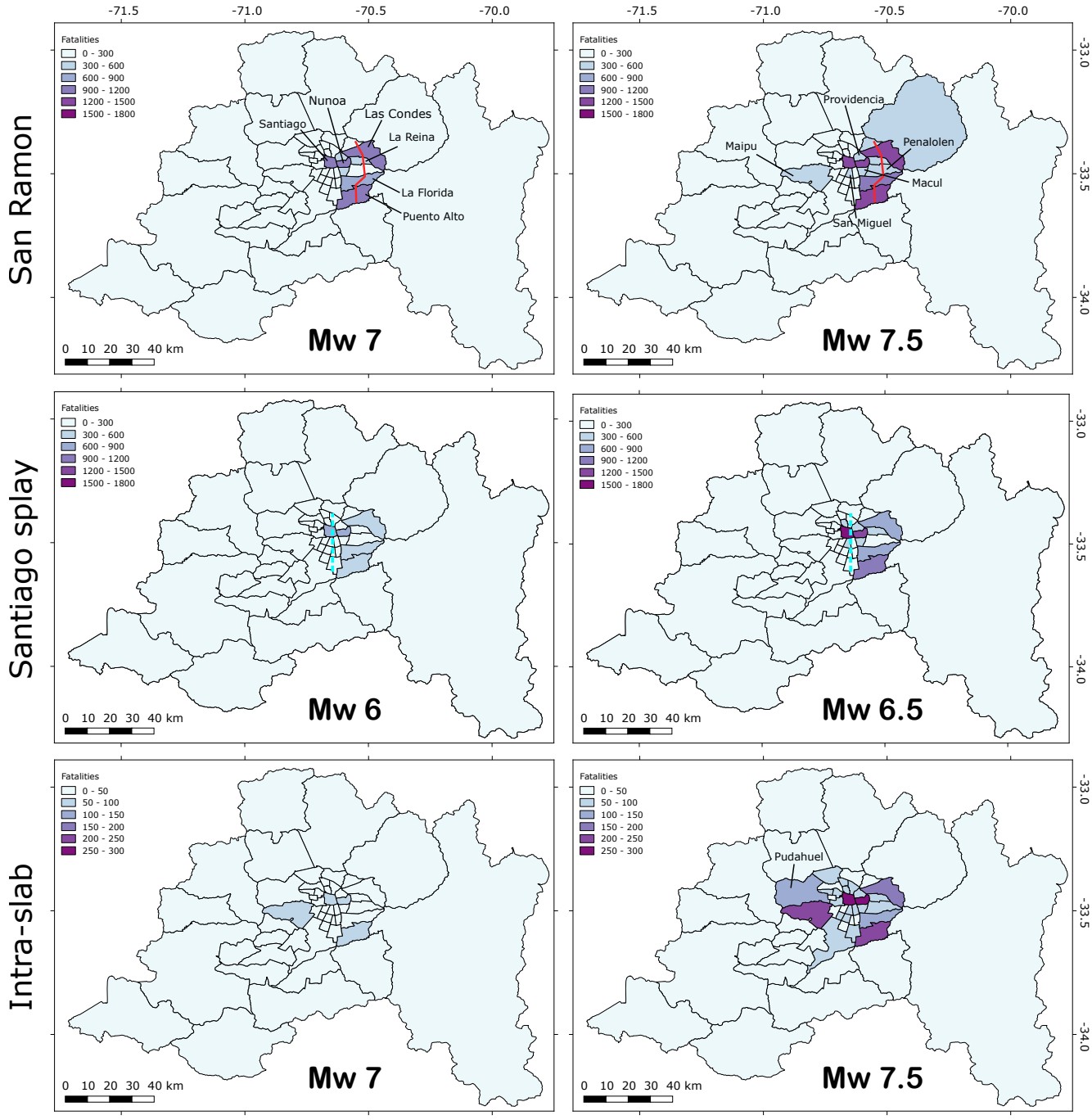

**Figure 12.** The estimated fatalities in each comuna for the earthquakes considered in each scenario for the San Ramón Fault (red line), the Santiago splay fault (dashed cyan line) and a deep intra-slab fault. Note that the range of the colour scale changes between different upper four and lower two panels.

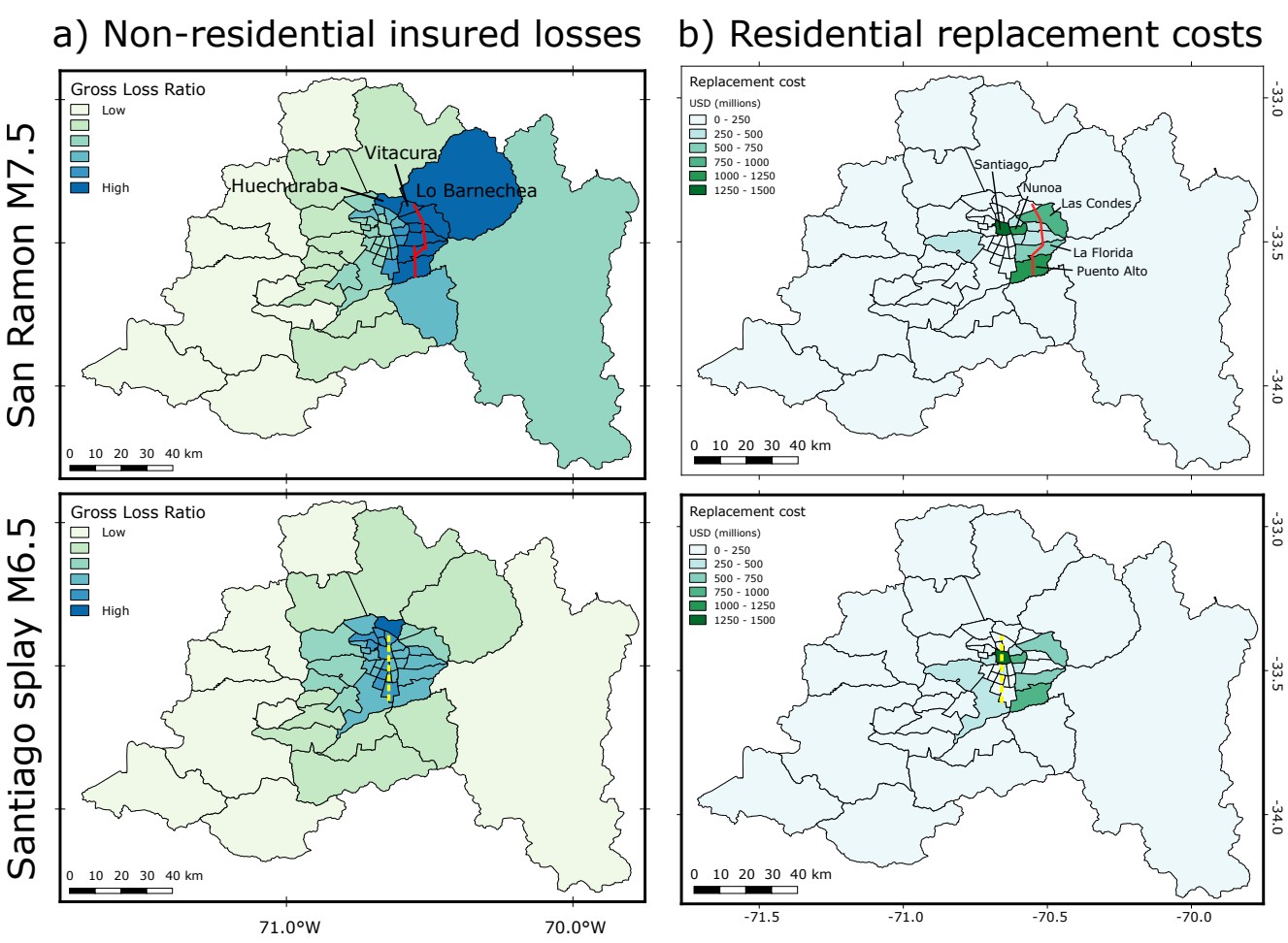

**Figure 13.** The distribution of (a) non-residential, and (b) residential replacement costs for maximum magnitude scenarios considered for the San Ramón Fault and the Santiago splay fault (moment magnitude 7.5 and 6.5 respectively). The gross loss ratios represent the calculated losses in each scenario over the total insured value after the application of policy conditions and deductibles. The residential replacement costs are the costs to repair/replace buildings and their contents damaged in each scenario. The residential replacement cost maps for all scenarios are given in Fig. S8.

**Table 1.** Summary of the building classes and typologies in our Santiago residential building exposure model, based on Yepes-Estrada et al. (2017).

| Class | Building typologies (GEM taxonomy) | Count | % of total | Storeys |
|---|---|---|---|---|
| RC | Non-ductile reinforced concrete walls (CR_LWAL-DNO) | 79,198 | 5.65 | 1–7 |
| | Ductile reinforced concrete walls (CR_LWAL-DUH) | 72,882 | 5.20 | 1–8+ |
| MCF | Non-ductile confined masonry walls (MCF_LWAL-DNO) | 395,349 | 28.19 | 1–3 |
| | Ductile confined masonry (MCF_LWAL-DUH) | 149,714 | 10.68 | 1–5 |
| MR | Ductile reinforced masonry walls (MR_LWAL-DNO) | 259,223 | 18.48 | 1–3 |
| | Non-ductile reinforced masonry (MR_LWAL-DUH) | 109,430 | 7.80 | 1–5 |
| MUR | Non-ductile unreinforced masonry walls (MUR_LWAL-DNO) | 161,779 | 11.54 | 1–2 |
| | Non-ductile unreinforced adobe walls (MUR-ADO_LWAL-DNO) | 32,160 | 2.29 | 1–2 |
| W | Non-ductile light wood walls (W-WLI_LWAL-DNO) | 12,796 | 0.91 | 1–2 |
| | Ductile light wood walls (W-WLI_LWAL-DUM) | 129,372 | 9.23 | 1–3 |
| UNK | Unknown or Insufficient information available (UNK) | 319 | 0.02 | |
| | **Total** | 1,402,222 | | |

**Table 2.** Exposed populations and buildings for the 10 most affected communes in terms of average modelled building collapse across the 6 earthquake scenarios; full list in supplementary material (Tables S2 and S3). The communes are ranked in order of average collapse count.

| Comuna | Area (km$^2$) | Population | Pop$^n$ density (per sq km) | Pop$^n$ below poverty line[a] (%) | Buildings in exposure model (% of Total) | | | | | |
|---|---|---|---|---|---|---|---|---|---|---|
| | | | | | RC | MCF | MR | MUR | W | Total |
| Puente Alto | 88 | 622,356 | 7,072 | 8.0 | 3 | 61 | 29 | 3 | 4 | 143,463 |
| La Florida | 71 | 356,925 | 5,027 | 3.1 | 6 | 51 | 26 | 9 | 8 | 81,493 |
| Santiago | 22 | 371,250 | 16,875 | 5.9 | 38 | 19 | 15 | 27 | 1 | 57,341 |
| Ñuñoa | 17 | 273,354 | 16,080 | 2.4 | 36 | 32 | 18 | 13 | 1 | 42,598 |
| Las Condes | 99 | 296,251 | 2,992 | 0.6 | 38 | 31 | 21 | 8 | 1 | 51,646 |
| Maipú | 133 | 608,094 | 4,572 | 5.2 | 4 | 44 | 35 | 12 | 6 | 142,828 |
| Peñalolén | 54 | 197,909 | 3,665 | 4.8 | 5 | 39 | 30 | 9 | 17 | 42,562 |
| La Pintana | 31 | 191,306 | 6,171 | 13.9 | 2 | 39 | 36 | 10 | 13 | 40,847 |
| Providencia | 14 | 88,928 | 6,352 | 0.7 | 51 | 25 | 18 | 6 | 0 | 22,080 |
| Macul | 13 | 116,694 | 8,976 | 5.3 | 16 | 32 | 17 | 27 | 7 | 23,528 |

[a] defined as $400 monthly income (in 2015 US dollars) for a family of 4 (Ministerio de Desarrollo Social, 2016)

**Table 3.** Gross loss (GR) ratios for the Santiago non-residential exposure.

| Event | | Average GR Loss Ratio (Loss/Total Insured Value) |
|---|---|---|
| **Fault** | **Magnitude** | |
| San Ramón | 7.0 | 5 % |
| San Ramón | 7.5 | 10 % |
| Santiago splay | 6.0 | 0 % |
| Santiago splay | 6.5 | 5 % |

**Table 4.** The fraction of collapsed residential buildings by building class (Table 1) normalised to the total collapse fraction in each earthquake scenario. SR - San Ramón, SS - Santiago Splay, IS - Intra-slab. The number indicates the moment magnitude of the earthquake source. Values greater than one indicate that the building typology is more likely to collapse than the average.

| Building typology | Total exposed | Normalised Collapsed Fraction | | | | | |
|---|---|---|---|---|---|---|---|
| | | SR7 | SR7.5 | SS6 | SS6.5 | IS7.0 | IS7.5 |
| RC | 152,080 | 0.7 | 0.7 | 0.6 | 0.7 | 0.1 | 0.2 |
| MCF | 545,063 | 0.9 | 0.9 | 0.8 | 0.9 | 0.8 | 0.9 |
| MR | 368,653 | 0.8 | 0.8 | 0.8 | 0.9 | 0.7 | 0.8 |
| MUR | 193,939 | 2.0 | 1.9 | 3.0 | 2.6 | 3.4 | 2.8 |
| W | 142,168 | 0.2 | 0.2 | 0.2 | 0.3 | 0.1 | 0.2 |
| **Total** | 1,401,903 | | | | | | |

**Table 5.** Summary of losses from each fault and earthquake scenario.

| Fault | Earthquake magnitude | Building collapse | Fatalities | Replacement cost (billions USD) |
|---|---|---|---|---|
| San Ramón | 7 | 136,400 | 9,670 | 8.3 |
| San Ramón | 7.5 | 181,300 | 12,650 | 10.4 |
| Santiago splay | 6 | 107,300 | 6,470 | 6.1 |
| Santiago splay | 6.5 | 172,200 | 11,430 | 9.6 |
| Intraslab | 7 | 23,500 | 930 | 1.2 |
| Intraslab | 7.5 | 63,500 | 3,180 | 3.1 |
| Maule | 8.8 | 9,800 | 510 | 3.1 |