# Peer review of "Contrasting seismic risk for Santiago, Chile, from near-field and distant earthquake sources"

_Natural Hazards and Earth System Sciences, 2019_

## Referee Comment (RC1) · Robin Lacassin (Referee) · 18 Mar 2019

The paper by E. Hussain et al. uses a deterministic approach to discuss the risk and potential losses at the capital city of Chile due to different possible earthquake sources. I found the paper interesting, useful, and worth to be published in NHESS. I do not have major general critics and will only make comments to help improving the quality of the paper. However I am not expert in risk modeling and cannot judge the technical quality of this aspect of the work. So I will mainly comment on the seismotectonic framework and also briefly on the way the final conclusions are presented and summarized.

Detailed comments and suggestions:

1/ At least two papers evaluating the hazard (using PGA analyses) due to the San

Ramón Fault are not mentioned (Pilz et al. 2011, Estay et al. 2016). They should be referenced and discussed. One of these papers (Pilz et al. 2011) discusses the effects of the Santiago basin structure on the seismic waves and PGA. Are these effects tested in the present study (it seems to me not) ? If not why ? And what would be the drawbacks not doing it ? This needs at least some discussion.

2/ The authors present new geomorphic analyses of the San Ramón Fault. Basically, their results confirm all Armijo et al. 2010 observations and conclusions. In this regard this part of the study would not be actually needed to implement the subsequent risk calculations (using Armijo et al. conclusions would give the same results). Don't miss my point however; I do not suggest to delete the geomorphic part of the paper: it's very good for Science to have confirmation by an independent study. But the authors should clearly state what I mention above and try not overselling their results.

3/ p.2 Lines 11: I don't understand this value of 43 mm/yr for the plate convergence. Zheng et al. is really NOT the appropriate reference. Mat be you took this value from a table in this paper? But in this case it is not the relative NZ-SA plate motion but a sort of NZ plate absolute motion. Admitted value for NZ-SA is between $\sim$ 6.5 and 7 cm/yr. At 33°S-71°W, it was 7.8 cm/yr for NUVEL1A, and has been lowered a little in more recent global plate models: 7.1 cm/yr in MORVEL2010, 6.5 cm/yr in ITRF2014. To compute such values you may use online UNAVCO plate motion calculator.

4/ p.2 L12: not sure James 1971 is the best reference (it is now more an "historical" one in which the Andes are interpreted as built by magmatic accretion, not by tectonics). You may for ex. cite more recent review papers like Oncken et al. 2006, Armijo et al. 2015, or others.

4/ p.2 L17: give a better reference than "USGS" - or expand this reference.

5/ p.2 L25-26: same remark about the plate convergence rate (see point 3). You may also mention here the compatibility between the long term rate estimated from balanced cross sections and the recurrence-slip characteristics deduced from the identified paleo-earthquakes (see discussion in Riesner et al. 2017).

6/ p.4 L9: do not cite Riesner et al. 2017 here. This paper is about the fold and thrust belt, not on the "surface expression" of the San Ramon fault scarp. Appropriate ref. here is Armijo et al. 2010. Alternatively you may cite Riesner et al. at p.5 L1.

7/ p.14 L14-16: writing in conclusions "6000-9000 fatalities" is extremely strong. Written without precaution it may generate anxiety and outrage in the population and even stakeholders. This result is strongly model-dependent and is also based on different assumptions (including seismotectonic and mechanical ones). Also, in this conclusion, there is no mention of the long recurrence times and related important uncertainties (OK, this is because you are not doing probabilistic evaluations, but this is important to help people understand the meaning of your results). From a risk communication viewpoint, it seems important to soften a little this conclusion while adding few more explanations and reminders of the uncertainties, including perhaps a rapid summary of epistemic uncertainties in the modeling approach and initial assumptions.

8/ Figures 8 and 9: these maps present the results as absolute values (number of building collapse or fatalities in each sector to the town). This seems less scientifically rigorous than using relative values (%) though I understand that giving such absolute values may be useful for people (stakeholders) using risk assessment. To allow easier quantitative comparison between the different cases, it would be wiser to use percentages that do not depend on the initial amount of building / people to be affected. This, plus the fact that the used scales are different from one case to the other, make comparisons of the different maps and of different sectors of the town quite difficult.

Robin Lacassin, Paris, March 2019.

Useful references :

Armijo, R., Lacassin, R., Coudurier-Curveur, A., Carrizo, D., Coupled tectonic evolution of Andean orogeny and global climate. (2015) Earth Science Reviews doi:

10.1016/j.earscirev.2015.01.005

Estay, N. P., Yáñez, G., Carretier, S., Lira, E., & Maringue, J. (2016). Seismic hazard in low slip rate crustal faults, estimating the characteristic event and the most hazardous zone: study case San Ramón Fault, in southern Andes. Natural Hazards & Earth System Sciences, 16 (12). doi:10.5194/nhess-16-2511-2016

Oncken, O., Hindle, D., Kley, J., Elger, K., Victor, P., and Schemmann, K. (2006). Deformation of the central andean upper plate system—facts, fiction, and constraints for plateau models. In The Andes, pages 3–27. Springer.

Pilz, M., Parolai, S., Stupazzini, M., Paolucci, R., and Zschau, J. (2011) Modelling basin effects on earthquake ground motion in the Santiago de Chile basin by a spectral element code: Geophysical Journal International, v. 187, p. 929– 945, doi:10.1111/j.1365-246X.2011.05183.x

Riesner, M., R. Lacassin, M. Simoes, R. Armijo, R. Rauld, and G. Vargas (2017), Kinematics of the active West Andean fold-and-thrust belt (Central Chile): structure and long-term shortening rate, Tectonics, 36, doi: 10.1002/2016TC004269.

UNAVCO plate motion calculator: https://www.unavco.org/software/geodetic-utilities/plate-motion-calculator/plate-motion-calculator.html

---

## Short Comment (SC1) · 27 Apr 2019

1. "There are many GMPEs available in the literature (see Douglas (2011)for a review)." -> Douglas (2011) does not provide a "review" but simply a compendium of published models. If you wish to reference this compendium I suggest you cite my website: www.gmpe.org.uk, which has the latest version, rather than this relatively old report. If you wished to cite a review article you could cite: Douglas and Edwards (Earth-Science Reviews, 2016, doi: 10.1016/j.earscirev.2016.07.005). 2. "We also take into account the local site effects by using the Vs30 values - shear wave velocity in the top 30 m of soil,estimated from a microzonation study by Bonnefoy-Claudet et al. (2009) (Fig. S4)." -> The location considered for your analysis is in a sedimentary basin with relatively soft soils ($\sim$300m/s according to S4), which are probably also deep ($\gg$50m).

Therefore, only using Vs30 to account for the effects of these sediments is probably underestimating the soil amplification as it generally ignores basin effects (e.g. Joyner, Bulletin of the Seismological Society of America, 2000). Some GMPEs roughly account for these effects (but not the ones you have chosen for this analysis). It may be worth checking the impact of accounting for these effects. 3. "We find that the fraction of reinforced concrete (RC) buildings in any commune decreases dramatically with the proportion of people living below the poverty line (Fig. 7)" -> Figure 7 shows that the scatter in the data for RC (and other building types) is large. It could be useful to check the statistical significance of the best-fit lines as it may be that the apparent trends are not robust.

---

## Author Comment (AC1) · 11 Jul 2019

Dear Dr Douglas,

We thank you for your comments and suggestions. Please find our responses below. Our response is in red.

*1. "There are many GMPEs available in the literature (see Douglas (2011)for a review)." -> Douglas (2011) does not provide a "review" but simply a compendium of published models. If you wish to reference this compendium I suggest you cite my website: www.gmpe.org.uk, which has the latest version, rather than this relatively old report. If you wished to cite a review article you could cite: Douglas and Edwards (EarthScience Reviews, 2016, doi: 10.1016/j.earscirev.2016.07.005).*
- Thank you for the comment. We have updated the reference to the ones suggested

*2. "We also take into account the local site effects by using the Vs30 values - shear wave velocity in the top 30 m of soil,estimated from a microzonation study by Bonnefoy-Claudet et al. (2009) (Fig. S4)." -> The location considered for your analysis is in a sedimentary basin with relatively soft soils (~300m/s according to S4), which are probably also deep (»50m). Therefore, only using Vs30 to account for the effects of these sediments is probably underestimating the soil amplification as it generally ignores basin effects (e.g. Joyner, Bulletin of the Seismological Society of America, 2000). Some GMPEs roughly account for these effects (but not the ones you have chosen for this analysis). It may be worth checking the impact of accounting for these effects.*
- Thank you for the comment. While in this study we are unable to account for the complexities of basin resonance and topography, we attempt to take into account the basin amplification effect in our ground motion calculations by using the Vs30 velocities - the shear wave velocity in the top 30 m of soil estimated from a microzonation study by Bonnefoy-Claudet et al. (2009). But we have noted that this will still probably underestimate the full basin effects and is a limitation of our study. We have added this to the end of section 3.2.

*3. "We find that the fraction of reinforced concrete (RC) buildings in any commune decreases dramatically with the proportion of people living below the poverty line (Fig. 7)" -> Figure 7 shows that the scatter in the data for RC (and other building types) is large. It could be useful to check the statistical significance of the best-fit lines as it may be that the apparent trends are not robust.*
- We have included the coefficient of determination values for each of the fitted lines through the building fraction against poverty data. While it is clear that the the scatter in the masonry building data means the fit is poorly constrained. However the correlation for the RC and W buildings is significant, and shows a decreasing trend between the fraction of RC structures with increasing levels of poverty with the opposite trend for W structures.

---

## Author Comment (AC2) · 11 Jul 2019

*The paper by E. Hussain et al. uses a deterministic approach to discuss the risk and potential losses at the capital city of Chile due to different possible earthquake sources. I found the paper interesting, useful, and worth to be published in NHESS. I do not have major general critics and will only make comments to help improving the quality of the paper. However I am not expert in risk modeling and cannot judge the technical quality of this aspect of the work. So I will mainly comment on the seismotectonic framework and also briefly on the way the final conclusions are presented and summarized. Detailed comments and suggestions:*

*1/ At least two papers evaluating the hazard (using PGA analyses) due to the San Ramón Fault are not mentioned (Pilz et al. 2011, Estay et al. 2016). They should be referenced and discussed. One of these papers (Pilz et al. 2011) discusses the effects of the Santiago basin structure on the seismic waves and PGA. Are these effects tested in the present study (it seems to me not) ? If not why ? And what would be the drawbacks not doing it ? This needs at least some discussion.*

- Thank you for highlighting these important papers. We have included the results from the two papers in our revised manuscript. We now make reference to Estay et al., 2016 in our introduction. The basin amplification effect suggested by Pilz et al. 2011 is partly included in our calculations through the use of the VS30 velocities. But this alone will not include the full basin effects including those from resonance within the basin, which we are unable to account for in this study. We have added to the end of section 3.2, a statement on the numerical simulations by Pilz et al., (2011) and the fact that the basin is likely to amplify the PGV.  And noted that not accommodating for the basin resonance will mean that out models do not take into account the particular vulnerability of buildings of certain height prone to resonance, which was an important factor for example in the Kathmandu rupture and basin amplification in Nepal with resonance at 4-5 seconds (Galetzer et al., 2015).

*2/ The authors present new geomorphic analyses of the San Ramón Fault. Basically, their results confirm all Armijo et al. 2010 observations and conclusions. In this regard this part of the study would not be actually needed to implement the subsequent risk calculations (using Armijo et al. conclusions would give the same results). Don't miss my point however; I do not suggest to delete the geomorphic part of the paper: it's very good for Science to have confirmation by an independent study. But the authors should clearly state what I mention above and try not overselling their results.*

- Thank you for the comment. You are correct that our work confirms the results found by Armijo et al. 2010 for the central and northern part of the fault and we have made that clear in the manuscript. Additionally our work in this paper does extend the fault trace further to the south from that mapped by Armijo et al.

*3/ p.2 Lines 11: I don't understand this value of 43 mm/yr for the plate convergence. Zheng et al. is really NOT the appropriate reference. Mat be you took this value from a table in this paper? But in this case it is not the relative NZ-SA plate motion but a sort of NZ plate absolute motion. Admitted value for NZ-SA is between ∼ 6.5 and 7 cm/yr. At 33∘S-71∘W, it was 7.8 cm/yr for NUVEL1A, and has been lowered a little in more recent global plate models: 7.1 cm/yr in MORVEL2010, 6.5 cm/yr in ITRF2014. To compute such values you may use online UNAVCO plate motion calculator.*

- We thank the reviewer for pointing out this error. We have now changed the text and Figure 2 to use the MORVEL2010 relative plate velocities (7.5 cm/yr, calculated using the UNAVCO plate motion calculator).

*4/ p.2 L12: not sure James 1971 is the best reference (it is now more an "historical" one
in which the Andes are interpreted as built by magmatic accretion, not by tectonics).
You may for ex. cite more recent review papers like Oncken et al. 2006, Armijo et al.
2015, or others.*
- We have changed the citation to the more recent papers as suggested

*4/ p.2 L17: give a better reference than "USGS" - or expand this reference.*
- We have expanded the citation as suggested

*5/ p.2 L25-26: same remark about the plate convergence rate (see point 3). You may
also mention here the compatibility between the long term rate estimated from balanced cross sections
and the recurrence-slip characteristics deduced from the identified paleo-earthquasdskes (see
discussion in Riesner et al. 2017).*
- We have corrected the convergence rate and mentioned the similarity between the various estimates of
the slip rate on the San Ramon Fault.

*6/ p.4 L9: do not cite Riesner et al. 2017 here. This paper is about the fold and thrust
belt, not on the "surface expression" of the San Ramon fault scarp. Appropriate ref.
here is Armijo et al. 2010. Alternatively you may cite Riesner et al. at p.5 L1.*
- We have corrected the citations as suggested

*7/ p.14 L14-16: writing in conclusions "6000-9000 fatalities" is extremely strong. Written
without precaution it may generate anxiety and outrage in the population and even
stakeholders. This result is strongly model-dependent and is also based on different
assumptions (including seismotectonic and mechanical ones). Also, in this conclusion,
there is no mention of the long recurrence times and related important uncertainties
(OK, this is because you are not doing probabilistic evaluations, but this is important
to help people understand the meaning of your results). From a risk communication
viewpoint, it seems important to soften a little this conclusion while adding few more
explanations and reminders of the uncertainties, including perhaps a rapid summary of
epistemic uncertainties in the modeling approach and initial assumptions.*
- We have amended the conclusion to be less dramatic and included text explaining that these numbers
are subject to uncertainty arising from the various model inputs.

*8/ Figures 8 and 9: these maps present the results as absolute values (number of
building collapse or fatalities in each sector to the town). This seems less scientifically
rigorous than using relative values (%) though I understand that giving such absolute
values may be useful for people (stakeholders) using risk assessment. To allow easier
quantitative comparison between the different cases, it would be wiser to use percentages that do not
depend on the initial amount of building / people to be affected. This,
plus the fact that the used scales are different from one case to the other, make comparisons of the
different maps and of different sectors of the town quite difficult.*
- We have included maps of the loss fractions for the residential collapse, fatalities and replacement
costs in the supplement (Figures S7, S8 and S9 respectively) and in the text used both the model results
and fractional values when discussing the losses.

*Robin Lacassin, Paris, March 2019.*

*Useful references :*
*Armijo, R., Lacassin, R., Coudurier-Curveur, A., Carrizo, D., Coupled tectonic evolution of Andean*
*orogeny and global climate. (2015) Earth Science Reviews doi:10.1016/j.earscirev.2015.01.005*
*Estay, N. P., Yáñez, G., Carretier, S., Lira, E., & Maringue, J. (2016). Seismic hazard in*
*low slip rate crustal faults, estimating the characteristic event and the most hazardous*
*zone: study case San Ramón Fault, in southern Andes. Natural Hazards & Earth*
*System Sciences, 16 (12). doi:10.5194/nhess-16-2511-2016*
*Oncken, O., Hindle, D., Kley, J., Elger, K., Victor, P., and Schemmann, K. (2006). Deformation of the*
*central andean upper plate systemâ˘Tfacts, fiction, and constraints for ˘*
*plateau models. In The Andes, pages 327. Springer.*
*Pilz, M., Parolai, S., Stupazzini, M., Paolucci, R., and Zschau, J. (2011) Modelling basin*
*effects on earthquake ground motion in the Santiago de Chile basin by a spectral element code:*
*Geophysical Journal International, v. 187, p. 929 945, doi:10.1111/j.1365-*
*246X.2011.05183.x*
*Riesner, M., R. Lacassin, M. Simoes, R. Armijo, R. Rauld, and G. Vargas (2017),*
*Kinematics of the active West Andean fold-and-thrust belt (Central Chile): structure*
*and long-term shortening rate, Tectonics, 36, doi: 10.1002/2016TC004269.*
*UNAVCO plate motion calculator:*
*https://www.unavco.org/software/geodeticutilities/plate-motion-calculator/plate-motion-calculator.html*

References:
Galetzka, J., Melgar, D., Genrich, J.F., Geng, J., Owen, S., Lindsey, E.O., Xu, X., Bock, Y., Avouac,
J.P., Adhikari, L.B. and Upreti, B.N., 2015. Slip pulse and resonance of the Kathmandu basin during
the 2015 Gorkha earthquake, Nepal. Science, 349(6252), pp.1091-1095.

---

## Short Comment (SC2) · 8 Oct 2019

An important topic was analyzed in this work and can be a significant scientific contribution. Some comments are explained here, with the aim of improve the analysis and discussions of this work.

Major comments:

1. Lateral (N-S direction) continuity of the SRF

An important discussion has been obviated: the lateral-continuity (in a N-S direction) of the vertical offset of the fault, and which implications have in the fault rupture length. For example, (c) and (d) profile in Figure 4 do not have a defined offset to argue the fault presence. Additionally, differences in the offset ranging between 119 m and 5 m,

imply an unclear variability of the fault scarp. Does these offset represent the same geological process? Specifically, these offset also varies in the short-distance, for example varies between 30 and 2 meters in profiles (n) and (o) separated by only $\sim$ 1 km; or varies between 119 and 36 meters for profiles (h) and (i) separated only $\sim$1km. Thus, any obviously lateral-continuity is observed.

Then, (1.1) which criteria define a continuously fault? and (1.2) which criteria define a fault offset?

Additionally, Figure 4 does not have a descriptive caption to understand which represent the different colors of the lines (red, green and blue). (1.3) Please improve the caption of this figure.

(1.4) In the same Figure 4; Why red and green lines have different length including different number of points to make the linear regression? i.e. in profile (e), and (i) the green line starts in 0 distance, but in other profiles arbitrarily start at different distance.

(1.5) If one of the main goals of this work is define the fault trace: please discuss (1) lateral continuity, (2) the meaning of the fault surface expression and (3) comparison with the inferred sub-surface fault.

2. Peak Ground acceleration results

The PGA values are extremely important for this work, and must be presented, analyzed and discussed in the main manuscript. Please discuss the result of the PGA-values and presented it in the main manuscript.

From these results, arise the following questions:

(2.1) Why the PGA results of the SRF (upper panels in Figure S5) concentrated the higher values in a E-W elongated shape?

(2.2) Please use the same PGA-scale for all the earthquakes scenarios (Figure S5), to allow the comparison between them.

(2.3) Please add the USGS PGA map of the Mw=8.8 Maule earthquake.

3. Applications of the fragility curve

Fragility curve of Villar-Vega et al. (2017) may not represent the appropriate function for the earthquake scenarios of this work. First, Villar-Vega et al. calculated its fragility curves with 300 seismic records mainly composed by subduction earthquakes. Local crustal earthquakes mean a different frequency range, time of expose and spectral response. Thus, crustal events need an unique and independent analysis to created fragility curves. Additionally, differences between Chilean, Argentinean, Ecuadorian, Colombian, and Peruvian seismic regulations for building need careful attention. "Despite the usefulness of these models, it is important to acknowledge their limitations and range of applicability. These fragility functions do not capture the specific features of the building stock at the local level. For the assessment of earthquake losses at a local scale, models derived using a more detailed methodology and considering the local characteristics of the building stock should be considered." (Villar-Vega et al. 2017).

(3.1) Please discuss the applicability of the fragility curve to the Santiago building stock.

(3.2) Note that in supplementary figure S6. Fragility curve are referenced to Villar-Vega 2014.

(3.3) How to explain losses and fatalities for earthquakes scenarios, if the Bessason et al. (2012) vulnerability curve only expose losses with PGA up to 1 g; and according to yours models the major PGA values are 0.8 g (Figure S5)?

(3.4) Why do you present a MMI curve (Naguit et al. 2017) if your results are explained in terms of the PGA values?

(3.5) How can you explain that the mean building collapse expected for the Mw=8.8 earthquake according to your results, it is overestimated 200% above of the observed damage?

(3.6) Can you compare the results of the Maule, 2010 earthquake in terms of the

fatalities?

4. Conclusions and uncertain of this work

Section 5.4 discuss the limitations and uncertain of this work. However, these discussions are not explicit in the abstract and conclusion sections. To provide the limitation of this work, please add these uncertain at least in the conclusion section.

Minor comments:

Page 5 Paragraph between line 13 and 20: The 1647 earthquake can be attributed to several sources, one of them is the splay fault beneath Santiago; but there are other options; (1) intraplate earthquake as the 1939 Chillan Earthquake; taking into account the ∼200 second of duration (ref libro); (2) another crustal fault next to Santiago, for example the West Andean Thrust faults (Armijo et al. 2010) or other fault describe by other authors (e.g. Farias et al. 2010). Further efforts must be done to state this paragraph, and any of these possibilities can be obviated and not mentioned.

Page 9 line 12: "for the commune of Las Condes"

Page 11 Line 10: the reference is Vargas et al. 2018, not Easton et al. 2018.

Page 12: Line 12: Please add "According our models, it is clear that there is a trade-off"

---

## Referee Comment (RC2) · Anonymous Referee #2 · 18 Dec 2019

The paper compares potential impacts of near field and distant earthquakes. The results show the need to take into account minor but proximal faults when addressing seismic risks in urban areas. In my opinion the paper is of good quality and may interest many readers. I suggest to accept it with the following revisions.

1) I'm not sure that the large impact of local/crustal earthquakes compared to subduction earthquakes is a new result. Similar conclusions have been reached in the 90s during the Earthquake Risk Management of the Quito city (Chatelain, J., Tucker, B., Guillier, B. et al. Earthquake risk management pilot project in Quito, Ecuador. Geo-Journal 49, 185–196 (1999) doi:10.1023/A:1007079403225). Other references may exist.

2). Are the selected GMPE's similar to the one selected by the SARA project ? If not,

why ?

3) The exposure model is "aggregated" at the census district level. The active faults are however "near field" and close to faults the ground-shaking intensity is highly distance dependent. The authors should better describe the distance computation between the faults and the exposure assets. Which distance definition is used ? Is this distance definition similar for all GMPE's ?

4) My understanding is that the building are homogenously distributed within the district which is not the case in reality. Such spatial homogenization may introduce a bias. The author should discuss the potential effects of this homogenization (and even test it using various random buildings distributions in each district cell)

5) The authors used the vs30 values from the Bonnefoy et al. (2009). This paper is however not deriving such vs30 values but resonant frequencies from H/V values. I then do not understand how vs30 values have been obtained.

6) Epistemic uncertainties are large for such risk computations. Such epistemic un-certainy is taking into account only for the GMPE part (for which several GMPE's are considered). The resulting uncertainty is however never shown in the paper (since the authors consider an average GMPE model). It would be interesting to show (on Figure 10) the results for each GMPE (and not the average) to illustrate (at least once) the impact of the modelling epistemic uncertainty on the results.

---

## Author Comment (AC3) · 28 Jan 2020

An important topic was analyzed in this work and can be a significant scientific contribution. Some comments are explained here, with the aim of improve the analysis and discussions of this work.

We would like to thank the Dr Estay for his instructive feedback and comments regarding our manuscript. Below we explain below how we have addressed each of the issues raised.

Major comments:
1. Lateral (N-S direction) continuity of the SRF
An important discussion has been obviated: the lateral-continuity (in a N-S direction) of the vertical offset of the fault, and which implications have in the fault rupture length. For example, (c) and (d) profile in Figure 4 do not have a defined offset to argue the fault presence. Additionally, differences in the offset ranging between 119 m and 5 m, imply an unclear variability of the fault scarp. Does these offset represent the same geological process? Specifically, these offset also varies in the short-distance, for example varies between 30 and 2 meters in profiles (n) and (o) separated by only ∼ 1 km; or varies between 119 and 36 meters for profiles (h) and (i) separated only ∼1km. Thus, any obviously lateral-continuity is observed.

Then,
(1.1) which criteria define a continuously fault? and (1.2) which criteria define a fault offset? Additionally, Figure 4 does not have a descriptive caption to understand which represent the different colors of the lines (red, green and blue). (1.3) Please improve the caption of this figure.
- Thank you for raising this important point. We are interested in segmentation only at a relatively large scale, and assume that there is continuity of the fault (or at least a lateral length of it) at depth given the overall geomorphic expression of the San Ramon mountains. Also the downsampling of the exposure to 1km, and the simplification of the GMPEs in terms of the distance to the fault, does not warrant trying to replicate the along strike variability of the fault too finely.

However, we agree that it is often difficult to determine the continuity of fault segments from geomorphological observations. This difficulty is compounded by ~8k years of erosion, slope degradation and the growth of folds as well as the urban expansion. The fault offset is estimated from the vertical offset between the best fit lines through the point cloud either side of the fault scarp. While these observations enable us to determine the active fault segments that comprise the San Ramon Fault, the variations in scarp height mean it is difficult to trace specific historical ruptures along the fault.

As discussed in the text, profile (d) shows no clear offset probably because of slope degradation due to the Mapocho river. We have included additional text in Section 2 of the manuscript discussing the issue of scarp continuity along-strike of the fault.

We have added additional information to the caption of Fig. 4 to improve clarity as suggested.

(1.4) In the same Figure 4; Why red and green lines have different length including different number of points to make the linear regression? i.e. in profile (e), and (i) the green line starts in 0 distance, but in other profiles arbitrarily start at different distance.
- The variable topographic slope along the fault means it is difficult to fit lines of equal length through the point clouds for each profile. In most cases we have tried to ensure a fit through at least 500m, but where possible ~1km, of points either side of the fault scarp. We have added this clarification in the text.

(1.5) If one of the main goals of this work is define the fault trace: please discuss (1) lateral continuity,

(2) the meaning of the fault surface expression and (3) comparison with the inferred sub-surface fault.
- We have included additional text in Section 2 of the paper discussing the issue of fault segmentation and our justification for using earthquake scenarios in our model that rupture across both main strands of the San Ramon Fault.

2. Peak Ground acceleration results
The PGA values are extremely important for this work, and must be presented, analyzed and discussed in the main manuscript. Please discuss the result of the PGA values and presented it in the main manuscript.

From these results, arise the following questions:
(2.1) Why the PGA results of the SRF (upper panels in Figure S5) concentrated the higher values in a E-W elongated shape?
(2.2) Please use the same PGA-scale for all the earthquakes scenarios (Figure S5), to allow the comparison between them.
(2.3) Please add the USGS PGA map of the Mw=8.8 Maule earthquake.
- (2.1) The shape of the San Ramon ground motion was due to an error in the way the OpenQuake engine calculated rupures in segmented faults. We have now corrected this and made the necessary amendments  to the manuscript. We have also included a figure of the ground motions in the main manuscript (Fig. 8) and additional text in section 3.4.
- (2.2) As suggested we have changed the ground motion figures to use the same colour scale for the San Ramon, Santiago splay and Maule scenarios. However if we set the colour scale to the largest ground motions (San Ramon 7.5), the spatial variation in ground shaking is not clearly visible for the intrslab case. Therefore while we have changed the colour scale to be same for all panels in Fig. 8 in the main manuscript we have decided to keep the intraslab colour scale in the supplements.
decided to leave the colour scale for the intraslab case as they are for .
- (2.3) We have included an additional figure in the manuscript  (Fig. 8) showing the ground motions for the largest earthquakes on each fault and the USGS PGA map for the Mw8.8 Maule earthquake.

3. Applications of the fragility curve
Fragility curve of Villar-Vega et al. (2017) may not represent the appropriate function for the earthquake scenarios of this work. First, Villar-Vega et al. calculated its fragility curves with 300 seismic records mainly composed by subduction earthquakes. Local crustal earthquakes mean a different frequency range, time of expose and spectral response. Thus, crustal events need an unique and independent analysis to created fragility curves. Additionally, differences between Chilean, Argentinean, Ecuadorian, Colombian, and Peruvian seismic regulations for building need careful attention. "Despite the usefulness of these models, it is important to acknowledge their limitations and range of applicability. These fragility functions do not capture the specific features of the building stock at the local level. For the assessment of earthquake losses at a local scale, models derived using a more detailed methodology and considering the local characteristics of the building stock should be considered." (Villar-Vega et al. 2017).
(3.1) Please discuss the applicability of the fragility curve to the Santiago building stock.
- Thank you for raising this important point. The methodology Villar-Vega et al. (2017) used for the fragility function analysis was to approximate each building typology as a single degree of freedom oscillator. So it is indeed correct that this method heavily simplifies the response of a building to ground shaking (and also one of the reasons why the uncertainties on the loss results are quite large). This approach would not work on a building-by-building scale analysis as mentioned in their paper.

However, we believe it is still sufficient to approximate the broader scale losses on a city district scale, which is why we show results summed over each district even though the initial calculations are done on a 1km x 1km grid.

We have included additional clarification of this point in the manuscript in section 3.3.

(3.2) Note that in supplementary figure S6. Fragility curve are referenced to Villar-Vega 2014.
 - This has now been corrected
(3.3) How to explain losses and fatalities for earthquakes scenarios, if the Bessason et al. (2012) vulnerability curve only expose losses with PGA up to 1 g; and according to yours models the major PGA values are 0.8 g (Figure S5)? (3.4) Why do you present a MMI curve (Naguit et al. 2017) if your results are explained in terms of the PGA values?
- Vulnerability functions are generally empirically derived using loss data, usually collected through insurance claims or governmental reports. Therefore these are necessarily approximated from one region of similar tectonic setting to another. The OpenQuake-engine contains a database of these functions and automatically uses the most appropriate functions for the defined setting. Figs. S6e and S6f were representative examples shown for illustrative purposes only. We have replaced these more appropriate examples used in our calculations.

(3.5) How can you explain that the mean building collapse expected for the Mw=8.8 earthquake according to your results, it is overestimated 200% above of the observed damage?
- As explained in section 3.4 of the manuscript, 4,306 collapsed buildings were recorded in the Santiago Metropolitan region in the Maule earthquake. While the collapse count is smaller than our modelled estimate of $9,800 \pm 8,000$, it is within the error margin. The discrepancy could have arisen due to a slightly different exposure model. The actual exposure in 2010 would have been different than our exposure model estimates, which uses data from 2014. Moreover, there is often ambiguity regarding the classification and reporting of actual structural collapse and damage beyond repair (and thus in need of demolition). See Villar-Vega et al. (2017a) for a discussion of this topic.

(3.6) Can you compare the results of the Maule, 2010 earthquake in terms of the fatalities?
- While fatality estimates exist for the toal losses in the Maule earthquake we have not been able to find reported values for the Santiago Metropolitan region alone.

4. Conclusions and uncertain of this work
Section 5.4 discuss the limitations and uncertain of this work. However, these discussions are not explicit in the abstract and conclusion sections. To provide the limitation of this work, please add these uncertain at least in the conclusion section.
- We have expanded the conclusion section of the manuscript to also include an acknowledgement of the uncertainties in our model calculations.

Minor comments:
Page 5 Paragraph between line 13 and 20: The 1647 earthquake can be attributed to several sources, one of them is the splay fault beneath Santiago; but there are other options; (1) intraplate earthquake as the 1939 Chillan Earthquake; taking into account the ∼200 second of duration (ref libro); (2) another crustal fault next to Santiago, for example the West Andean Thrust faults (Armijo et al. 2010) or other fault describe by other authors (e.g. Farias et al. 2010). Further efforts must be done to state this paragraph, and any of these possibilities can be obviated and not mentioned.
- Thank you for these suggestions. We have expanded this paragraph to make clear these other

possibilities for the 1647 earthquake and our justifications for choosing the buried splay and/or the deep intraslab as likely candidates.

Page 9 line 12: "for the commune of Las Condes"
- Corrected

Page 11 Line 10: the reference is Vargas et al. 2018, not Easton et al. 2018.
- Corrected

Page 12: Line 12: Please add "According our models, it is clear that there is a trade-off"
- Amended as suggested

---

## Author Comment (AC4) · 28 Jan 2020

The paper compares potential impacts of near field and distant earthquakes. The results show the need to take into account minor but proximal faults when addressing seismic risks in urban areas. In my opinion the paper is of good quality and may interest many readers. I suggest to accept it with the following revisions.

We would like to thank the reviewer for their instructive feedback and comments regarding our manuscript. Below we explain below how we have addressed each of the issues raised.

1) I'm not sure that the large impact of local/crustal earthquakes compared to subduction earthquakes is a new result. Similar conclusions have been reached in the 90s during the Earthquake Risk Management of the Quito city (Chatelain, J., Tucker, B., Guillier, B. et al. Earthquake risk management pilot project in Quito, Ecuador. GeoJournal 49, 185196 (1999) doi:10.1023/A:1007079403225). Other references may exist.
- Thank you for bringing this reference to our attention. While we agree that the local vs distal earthquake impact conclusion is not new, we are the first to show it using a scenario based earthquake risk analysis for the specific case of Santiago. We have included the highlighted reference in section 5.2 of the discussion to clarify this point.

2). Are the selected GMPE's similar to the one selected by the SARA project ? If not, why ?
- Yes, the GMPEs selected in this study are the same as the ones used in the SARA study as well as the study of Villar-Vega et al, 2017 (an output from the SARA project).

3) The exposure model is "aggregated" at the census district level. The active faults are however "near field" and close to faults the ground-shaking intensity is highly distance dependent. The authors should better describe the distance computation between the faults and the exposure assets. Which distance definition is used ? Is this distance definition similar for all GMPE's ?
- To remain consistent across the GMPEs we implement the form of the equations that uses the Joyner-Boore distances - defined as the shortest horizontal distance from each exposure element to the surface projection of the rupture area. We have clarified this point in section 3.2 of the manuscript.

4) My understanding is that the building are homogenously distributed within the district which is not the case in reality. Such spatial homogenization may introduce a bias. The author should discuss the potential effects of this homogenization (and even test it using various random buildings distributions in each district cell)
- The reviewer is correct that the buildings are homogenously distributed within each grid cell (1x1km). We believe this is sufficient for our study because in the end we aggregate up and present the results at the district level. Therefore any variations and/or bias introduced due to the equal distribution is minimised.

5) The authors used the vs30 values from the Bonnefoy et al. (2009). This paper is however not deriving such vs30 values but resonant frequencies from H/V values. I then do not understand how vs30 values have been obtained.
- Thank you for pointing out this error. The site study was initally conducted by Pasten 2007 for his Masters thesis where he made 264 measurements of the Vs30 for the local soils.

6) Epistemic uncertainties are large for such risk computations. Such epistemic uncertainy is taking into account only for the GMPE part (for which several GMPE's are

considered). The resulting uncertainty is however never shown in the paper (since the authors consider an average GMPE model). It would be interesting to show (on Figure 10) the results for each GMPE (and not the average) to illustrate (at least once) the impact of the modelling epistemic uncertainty on the results.

- Yes, we attempt to take into account the epistemic uncertainties in our modelling by using several GMPEs, which we average for our reported loss numbers. To avoid cluttering Fig. 10 in the manuscript we have included a similar figure but with the losses for each GMPE in supplementary Fig. S10.

---

## Author Response (AR2)

**Dear Dr Ulbrich,**
**Thank you for your comments. Please find below our response highlighted in red.**

…………………………………..

Dear Dr. Hussain and co-authors,
thank you again for your responses and the revised version. Based on the reviewers' comments and my own assessment of your responses, I request that you make some additional changes in order to come to the publication of your paper. While I appreciate that the reviewers' recommendation for just "technical corrections", I feel that there is still a potential issue in the vs30 values used, as pointed out by referee 2. Thus, following your next response and revision, I will either come to a decision myself, or have reviewer 2 have another look at this.

In detail, the vs30 map presented in Figure S4 is apparently showing a key dataset for your paper, but the origin is not clear. You assign the data source to Bonnefoy et al. (2009) in the caption, which cannot be correct according to reviewer 2, stating that Bonnefoy et al. (2009) was not deriving vs30 values but resonant frequencies from H/V values. In the text, you now assign the vs30 values used to a thesis by Pasten (Memoria para optar al grado de Magíster en Ciencias de la Ingeniería) which is apparently not accessible for readers of your manuscript. It is necessary that you clarify and describe the origin of the vs30 data unambiguously. If you took the data from Pasten's work, you must at least include a paragraph describing how and where they were obtained (in your response, you mention 264 measurements, but the current Figure S4 apparently contains a much larger number of points), and in how far they can be considered reliable. I suggest that the respective figure is moved from the supplement to the main text.
- The vs30 values used in our paper were indeed based on the microzonation work by Pasten [2007], and adapted by Leyton et al. [2011] and Humire-Guarachi [2013] who assigned Vs30 velocities to each zone using information from soil penetration tests at various sites. We have amended the citation to include a link to his report (http://repositorio.uchile.cl/handle/2250/102937).

In this paper two datasets were used to obtain the Vs30 information for Santiago. The first consists of local microzonation studies, which contain seismic zonation maps (Pasten, 2007; Leyton et al., 2011), and proposed Vs30 values for soil types in each zone taking into account additional information from soil penetration tests (Humire-Guarachi, 2013). However, given the cost and time demand of such studies, microzonation maps are usually focused on limited areas. Therefore, for the remaining zones we supplemented the micronzonation data using velocities from the USGS Global Vs30 Map Server (Allen and Wald, 2007). This method derives maps of seismic site conditions using topographic slope as a proxy, assuming that stiffer materials (i.e. higher Vs30 values) are more likely to maintain a steep slope, while deep basin sediments are deposited mainly in environments characterised by a lower velocity.

We have moved the Vs30 map into the main text into its own subsection (3.3), showing the Allen and Wald [2007] velocities sampled at the actual exposure sites, as well as indicated clearly which points are from the microzonation study.
Subsection: 3.3
Figure: 8

As an additional change, I request in accordance with the referee 2 assessment that you replace the Figure 11 in the paper with the current figure S10, of course adapting the caption accordingly.

- We have replaced Figure 11 as suggested.

With respect to referee 2's comment on the location of buildings you used, you explain the building exposure model in section 3.2 (with reference to figure S1), and mention the Joyner-Boore distance at page 8 line 32-33. You should, however, also mention explicitly that for the averaging of damage results into districts you apply weighting according to the spatial distribution of the number of buildings.

- Our district level losses are the sum of the losses of all points within that district, not the average. The loss calculations for each individual exposure point will include the distance weighting (Joyner-Boore). We explicitly state this in the main text.

[revised manuscript text omitted]